**COMMUNICATIONS**

# Single-cell lineage tracing by integrating CRISPR-Cas9 mutations with transcriptomic data

Hamim Zafar[1,2,3,5], Chieh Lin[4,5] & Ziv Bar-Joseph [3,4 ✉]

Recent studies combine two novel technologies, single-cell RNA-sequencing and CRISPR-Cas9 barcode editing for elucidating developmental lineages at the whole organism level. While these studies provided several insights, they face several computational challenges. First, lineages are reconstructed based on noisy and often saturated random mutation data. Additionally, due to the randomness of the mutations, lineages from multiple experiments cannot be combined to reconstruct a species-invariant lineage tree. To address these issues we developed a statistical method, LinTIMaT, which reconstructs cell lineages using a maximum-likelihood framework by integrating mutation and expression data. Our analysis shows that expression data helps resolve the ambiguities arising in when lineages are inferred based on mutations alone, while also enabling the integration of different individual lineages for the reconstruction of an invariant lineage tree. LinTIMaT lineages have better cell type coherence, improve the functional significance of gene sets and provide new insights on progenitors and differentiation pathways.

[1] Department of Computer Science and Engineering, Indian Institute of Technology Kanpur, Kanpur, India. [2] Department of Biological Sciences and Bioengineering, Indian Institute of Technology Kanpur, Kanpur, India. [3] Computational Biology Department, School of Computer Science, Carnegie Mellon University, Pittsburgh, PA, USA. [4] Machine Learning Department, School of Computer Science, Carnegie Mellon University, Pittsburgh, PA, USA. [5] These authors contributed equally: Hamim Zafar, Chieh Lin. ✉email: zivbj@cs.cmu.edu

  1

Reconstructing cell lineages that lead to the formation of tissues, organs, and complete organisms is of crucial importance in developmental biology. Elucidating the lineage relationships among the diverse cell types can provide key insights into the fundamental processes underlying normal tissue development as well as valuable information on what goes wrong in developmental diseases[1–3]. Traditionally, heritable markers have been utilized for prospective lineage tracing by first introducing them in a cell and then using them to track its descendants[3]. Such studies leveraged diverse markers, such as viral DNA barcodes[4], fluorescent proteins[5], mobile transposable elements[6], Cre-mediated tissue-specific recombination[7], and more. Other methods relied on retrospective lineage tracing by using naturally occurring somatic mutations[8,9], microsatellite repeats[10] or epigenetic markers[11]. While these approaches provided valuable insights, they are often limited to a small number of markers and cells and due to the lack of coupled gene expression information, they cannot characterize the diverse cellular identities of the tracked cells and their relation to the lineage branching[1].

Recent advances in single-cell transcriptomics (scRNA-seq) allow the profiling of thousands of individual cells and the identification of cell types at an unprecedented resolution[12–14]. Cost-efficient and scalable technologies provide large-scale scRNA-seq datasets that can be used to identify gene expression signatures of diverse cell types and to curate catalogs of cellular identities across tissues[13,15,16]. While some of these datasets have been used to infer developmental lineages[17], methods for such inference rely heavily on strong assumptions regarding expression coherence between developmental stages, which may not hold in all cases[17,18]. Moreover, these approaches alone are unable to recover intermediate cell types and states making it difficult to reconstruct the early developmental lineages in an adult organism[18,19].

Very recently, new experimental techniques that simultaneously recover transcriptomic profiles and genetic lineage markers from the same cell have been introduced[20–22]. One of the earliest methods using such approach is scGESTALT[20], which combines the CRISPR-Cas9-based lineage tracing method termed GESTALT[23] with droplet-based single-cell transcriptomic profiling. scGESTALT inserts Cas9-induced stochastic (random) mutations to a genomic CRISPR barcode array at multiple time points. The edited barcodes are then sequenced and used for reconstructing a lineage tree based on Maximum Parsimony (MP) criterion[24]. Cell types are independently inferred based on scRNA-seq data. Another method is ScarTrace[22], which utilizes identical target sites located on separate transgenes for introducing CRISPR-Cas9 mutations followed by SORT-seq sequencing to capture the transcriptome. Lineage trees are then reconstructed by using the MP principle on the mutation data.

While these and similar methods have been successfully applied to a number of organisms[20,21], they encompass several computational challenges. First, the random mutation data used for reconstructing the MP lineage is noisy and often saturated making it difficult to separate different cell types, especially at later stages. Even though expression information is collected for all genes in each cell, to date the reconstruction of the lineage tree solely depends on the stochastic Cas9-induced mutations. As a result, the resulting lineage tree sometimes fails to separate different types of cells and places similar cell types on distant branches. Further, multiple tree topologies can have the same parsimony score based on mutations making the reconstruction more challenging. In addition, the random nature of the induced mutations restricts the lineage reconstruction to each individual and mutation data from multiple individuals cannot be combined for inferring a single lineage tree based on multiple experiments.

To improve the reconstruction of lineages from CRISPR-Cas9 mutations and scRNA-seq data, we developed a statistical method, Lineage Tracing by Integrating Mutation and Transcriptomic data (LinTIMaT) that integrates mutational and transcriptomic data for reconstructing lineage trees in a maximum-likelihood framework. LinTIMaT employs a likelihood function for evaluating different tree structures based on mutation information. It then defines a likelihood optimization problem which combines the likelihood score for the mutation data with Bayesian hierarchical clustering[25], which evaluates the coherence of the expression information such that the resulting tree concurrently maximizes agreement for both transcriptomic data and genetic markers from the same cell. The tree space is explored by a heuristic search algorithm that first infers a lineage tree based on mutation information and further refines it based on both mutation and expression information. Finally, LinTiMaT also employs an algorithm for integrating lineages reconstructed for different individuals of the same species for inferring an invariant lineage tree. We applied LinTIMaT to both, simulated mutation data where ground truth is known and to zebrafish datasets generated using two different technologies[20,22]. As we show, by integrating transcriptomic and mutational data, LinTIMaT was able to improve the reconstruction of lineages when compared to MP method. In addition, we used LinTIMaT to combine data from multiple individuals for reconstructing an invariant lineage. As we show, such invariant lineage further improved on each of the individual lineages in terms of both clade homogeneity and functional assignment for the cells residing on the leaves of the lineage tree.

## Results

**Overview of LinTIMaT.** An overview of LinTIMaT is shown in Fig. 1. We assume that the cell lineage is a rooted directed tree (Fig. 1a), root of which denotes the initial cells that do not contain any marker (or editing event). The leaves of this tree denote the cells from which the mutated barcodes and RNA-seq data have been recovered. The CRISPR-Cas9 edits are acquired on the branches of the lineage tree as the single-cell zygote transforms into an adult organism. For expression data, the method assumes that cells under an internal node can either display similar expression profile (low variance owing to similar cell type) or two or more different expression profiles (high variance) if they later split into multiple cell types. Supplementary Fig. 1 shows the generative process assumed by LinTIMaT.

LinTIMaT reconstructs the lineage tree by maximizing a likelihood function that accounts for both mutation and expression data. The likelihood function imposes a Camin-Sokal parsimony criterion for each synthetic marker. The probability associated with a transition of mutation state for a marker along a branch of the lineage tree is computed based on the abundance of the marker in the single cells. To compute the expression likelihood based on the transcriptomic data, the lineage is modeled as a Bayesian hierarchical clustering (BHC)[25] of the cells and the marginal likelihoods of all the partitions consistent with the given lineage tree are computed based on a Dirichlet process mixture model. To optimize the tree topology, we employ a heuristic search algorithm, which stochastically explores the space of lineage trees.

The above algorithm reconstructs trees for a specific CRISPR-Cas9 mutation set. To integrate trees resulting from repeat experiments of the same organism, LinTIMaT further reconstructs a species-invariant lineage tree (Fig. 1b). Our model assumes that a subset of the lineages (and cells) are conserved between different individuals of the same species. Our invariant lineage reconstruction algorithm attempts to identify such

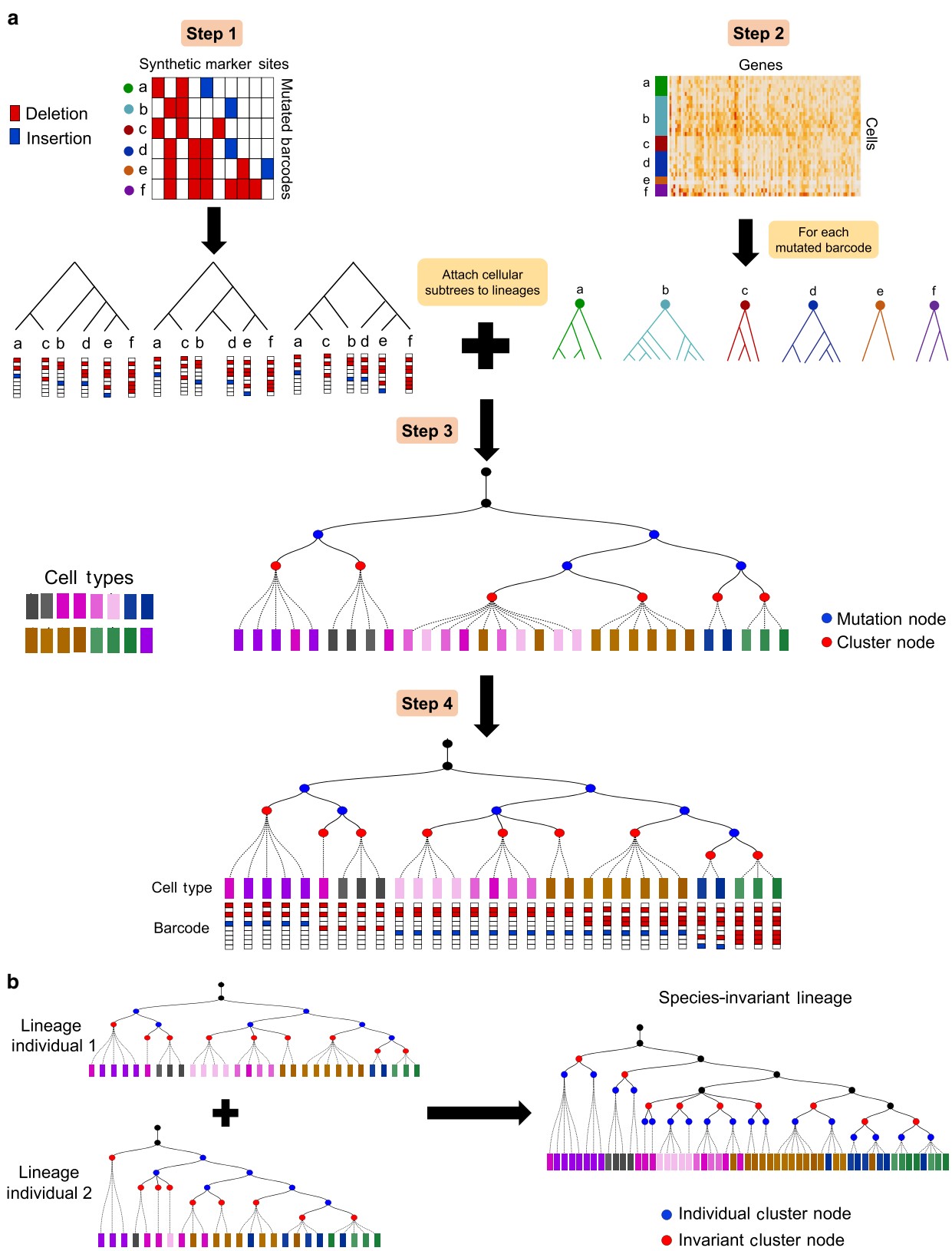

invariant groups of cells based on both, their similar expression pattern and their branching history. After determining the clusters that are preserved in all the individual lineages (based on expression similarity), the method starts with an initial greedy matching and iterates to minimize an objective function consisting of two distance functions, the first is aimed at minimizing the disagreement between the topology of the invariant lineage and the individual lineages while the second distance is minimized for improving the matching of the preserved cell clusters. See "Methods" for complete details.

**Fig. 1 Overview of LinTIMaT. a** LinTIMaT reconstructs a cell lineage tree by integrating CRISPR-Cas9 mutations and transcriptomic data. In Step 1, LinTIMaT infers top scoring lineage trees built on barcodes using only mutation likelihood. In Step 2, for all cells carrying the same barcode, LinTIMaT reconstructs a cellular subtree based on expression likelihood. In Step 3, cellular subtrees are attached to barcode lineages to obtain cell lineage trees and the tree with the best combined likelihood is selected. Finally, LinTIMaT uses a hill-climbing search for refining the cell lineage tree by optimizing the combined likelihood (Step 4). **b** To reconstruct a species-invariant lineage, LinTIMaT first identifies cell clusters that are preserved in all individual lineages and then performs an iterative search that attempts to minimize the distance between individual lineage trees and the invariant tree topology. As part of the iterative process, LinTIMaT matches preserved clusters in one individual tree to preserved clusters in other individual tree(s) such that leaves in the resulting invariant tree contain cells from all individual studies. See Methods for complete details.

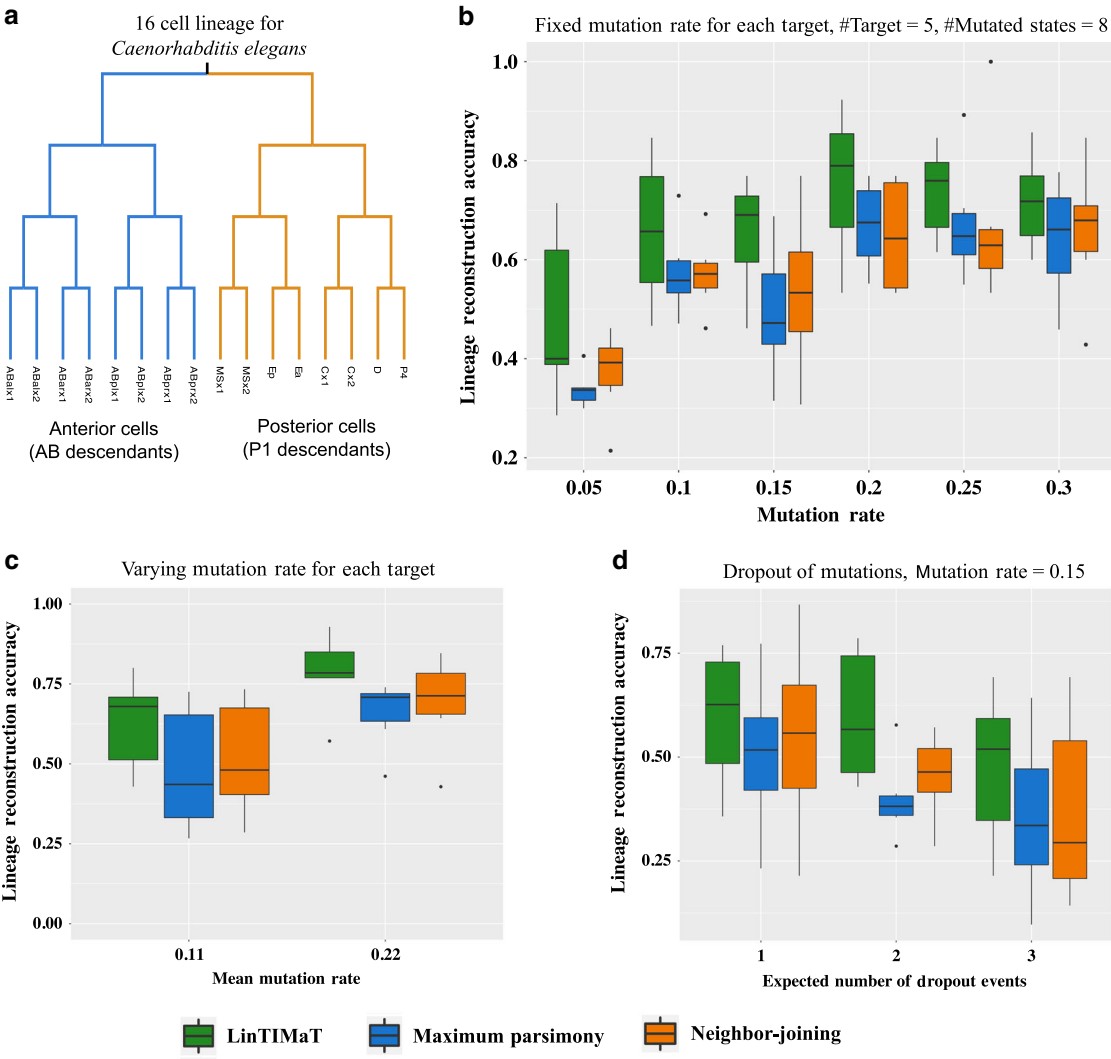

**Fig. 2 Benchmarking on C. elegans lineage. a** 16-cell embryo lineage for *Caenorhabditis elegans*. scRNA-seq data for each leaf (cell) was obtained from[27] and included 6 replicates for each cell. **b** Comparison of LinTIMaT, Camin-Sokal Maximum Parsimony, and Neighbor-joining when varying the mutation rates. The number of possible mutational states was set to 8. Fixed mutation rate was used for each CRISPR target. Each box plot summarizes results for six replicates with varying simulated CRISPR mutation data and experimental scRNA-seq data. **c** Comparing lineage reconstruction methods when mutation rate varies between different target sites. **d** Comparison of accuracy of lineage reconstruction by LinTIMaT, Camin-Sokal Maximum Parsimony, and Neighbor-joining in the presence of mutation dropout. Fixed mutation rate, $\mu = 0.15$ was used for all targets. For **b–d** each box-and-whisker plot summarizes results for six replicates, where the box shows the interquartile range (IQR, the range between the 25th and 75th percentile) with the median value, whiskers indicate the maximum and minimum value within 1.5 times the IQR, also shown are outliers as black dots.

**Benchmarking LinTIMaT using a *Caenorhabditis elegans* dataset**. We first tested whether the underlying assumptions LinTIMaT is based on, namely that gene-expression information can be used to reduce errors in mutation data for lineage reconstruction, actually hold. For this, we used a well-resolved exact lineage from *Caenorhabditis elegans*[26]. To benchmark LinTIMaT, we combined experimental *C. elegans* scRNA-seq data

with simulated CRISPR-Cas9 mutation data. scRNA-seq data was obtained from Tintori et al.[27] who profiled the 16-cell embryos of *C. elegans*. Since we know the lineage for these cells (Fig. 2a), we could use it to simulate CRISPR-Cas9 mutations based on the method proposed by Salvador–Martínez et al.[28]. Simulated datasets emulated potential errors encountered in a CRISPR-based lineage tracing experiment. These include issues related to

variability in the mutation rate ($\mu$) for each cell division, site-specific variability in mutation rates for different target sites and dropouts of CRISPR mutations which refer to erasing some of the earlier lineage mutations by later ones[29], see Methods for details. The lineage reconstruction performance of LinTIMaT on this *C. elegans* benchmark dataset was compared against that of the Camin-Sokal MP method, which was used on the scGESTALT dataset[20] for reconstructing lineage trees from CRISPR mutation data and the neighbor-joining (NJ) method for reconstructing phylogenetic trees[30]. The accuracy of lineage reconstruction was measured based on a metric used in[28] and Robinson-Foulds (RF) distance[31] between the true lineage tree and the inferred lineage tree (see Methods for details).

Figure 2b compares LinTIMaT, MP, and NJ for varying mutation rates. For all values of mutation rates, LinTIMaT achieved higher accuracy in lineage reconstruction compared to that of MP and NJ. For lower mutation rates ($\mu \leq 0.15$), LinTIMaT achieved upto 41.64% improvement in mean lineage reconstruction accuracy over that of MP, upto 29.45% improvement over that of NJ and lower RF distance compared to the FP and FN distances for the trees inferred by MP and NJ (Supplementary Fig. 2). Performance of MP and NJ improved with an increase in mutation rate but even for datasets with higher mutation rates ($\mu \geq 0.2$), LinTIMaT was able to achieve better lineage reconstruction accuracy (upto 12.9% and 16.57% improvements over MP and NJ respectively) (see Supplementary Discussion for more details). Next, on datasets with mutation rate differing between sites, LinTIMaT achieved higher accuracy (13.56–30.37% improvement, Fig. 2c) and lower RF distance compared to that of MP and NJ (Supplementary Fig. 3) indicating its robustness to higher complexity in the CRISPR mutational history. Erasure of some of the earlier lineage records (i.e. CRISPR mutation dropouts)[20] have been shown to have significant impact on the lineage reconstruction accuracy[28]. To analyze LinTIMaT's performance in the presence of dropouts, we simulated datasets with different dropout rates for a fixed mutation rate (expected number of dropout events in the cell lineage, $\epsilon_d = \{1, 2, 3\}$, $\mu = 0.15$). For all settings, LinTIMaT achieved better accuracy (Fig. 2d) and smaller RF distance (Supplementary Fig. 4) compared to MP and NJ (see Supplementary Discussion for details). LinTIMaT was also able to consistently obtain higher accuracy compared to MP and NJ for $\epsilon_d = 2$ and mutation rate varying from $\mu = 0.05$ to $\mu = 0.3$ (Supplementary Fig. 5).

**LinTIMaT recovers convergent and divergent differentiation**. During development, cells with distinct developmental history can converge onto the same mature cell type (convergent differentiation) while cells sharing close ancestry can differentiate into distinct transcriptional states (divergent differentiation)[32]. To assess LinTIMaT's ability to handle these scenarios, we combined zebrafish scRNA-seq data from scGESTALT[20] with synthetic CRISPR-Cas9 mutation data from simulated lineage of 100 cells containing convergent and divergent differentiation.

For convergent differentiation, we simulated cell lineages where forebrain neuron cells were divided into two groups and placed in two different subtrees (see Supplementary Discussion for details). We simulated lineages under two different ancestry settings for the cells (Supplementary Figs. 6a and 7a, Supplementary Discussion) and simulated CRISPR mutations with different dropout rates. For all experimental settings, LinTIMaT's lineage reconstruction error was lower when compared to the error resulting from placing the two groups in the same subtree (Supplementary Figs. 6b and 7b). Specifically, for all but 2 of the

30 simulations, LinTIMaT was able to correctly place the two groups in different subtrees (Supplementary Figs. 6c and 7c).

Next, we assessed LinTIMaT's ability to infer the lineage relationship between two groups of cells that underwent divergent differentiation (see Supplementary Fig. 8a, Supplementary Discussion for details). Again, LinTIMaT achieved lower lineage reconstruction error when compared to the error resulting from placing the two groups in different subtrees (Supplementary Fig. 8b) and in all cases correctly placed the two in the same subtree (Supplementary Fig. 8c).

**LinTIMaT improves lineage tree reconstruction**. Next, we applied LinTIMaT to analyze two experimental zebrafish datasets[20,22], each using a different technology for inserting CRISPR-Cas9 mutations. The first dataset was generated using scGESTALT[20]. The second dataset was generated using ScarTrace[22].

For the scGESTALT dataset, we applied LinTIMaT on two zebrafish samples, ZF1 and ZF3, consisting of 750 and 376 cells respectively, from which both the transcriptome (20287 genes) and edited barcode (192 unique barcodes, 324 unique markers for ZF1 and 150 unique barcodes, 265 unique markers for ZF3) were recovered. For both fishes, our analysis shows that improving the likelihood function used by LinTIMaT increases the coherence of the resulting cell types for each subtree, without impacting the overall mutation likelihood (Fig. 3a and Supplementary Fig. 9). For both fishes, LinTIMaT generated highly branched multiclade lineage trees (Fig. 3b and Supplementary Fig. 10). Blue nodes on the tree represent mutation events assigned while red nodes represent the clusters identified based on gene-expression data. It is important to note that cluster nodes do not necessarily represent common ancestors for the cells underneath, instead, cluster nodes are a way of grouping nearby cells together based on expression information without affecting the mutational ancestor-descendant relationships. ZF1 lineage tree comprised 25 major clades (level 1 tree nodes) and 113 cluster nodes, 77 of which consisted of more than one cell. ZF3 lineage tree comprised 17 major clades and 42 cluster nodes, 33 of which consisted of more than one cell. We compared the lineage trees reconstructed by LinTIMaT to the trees reconstructed using MP as used in the original study[20] by comparing the accuracy of cell clusters in the trees. In the original study, 63 transcriptionally distinct cell types were identified using an unsupervised, modularity-based clustering approach from 6 zebrafish samples. We used this clustering to compute the Adjusted Rand Index (ARI) for the cell clustering obtained from a lineage tree (Methods). For MP lineage trees, the unique barcodes represent cell clusters as mutation information was the only basis for reconstructing the tree. For each fish, the lineage tree reconstructed by LinTIMaT resulted in better cell clustering (37.5% and 36.4% improvement in ARI for ZF1 and ZF3 respectively) compared to MP results based on mutation data alone (see Supplementary Table 1 and Supplementary Discussion for details).

Lineage trees reconstructed using LinTIMaT showed successful integration of mutation and expression data. When using only mutation data, in several cases, cells belonging to very different cell types were clustered together. In contrast, in LinTIMaT reconstructed trees, these cells were correctly assigned to different subtrees corresponding to different cell types. Clade a1 in ZF3 lineage tree (Fig. 3c) is one such example. In MP lineage tree for ZF3, neural progenitor cells, hindbrain granule cells, and neurons in ventral forebrain and hypothalamus (total 43 cells) were clustered together under clade a1 as they shared the same mutational barcode. The tree reconstructed by LinTIMaT correctly separated these cells into three major subtrees

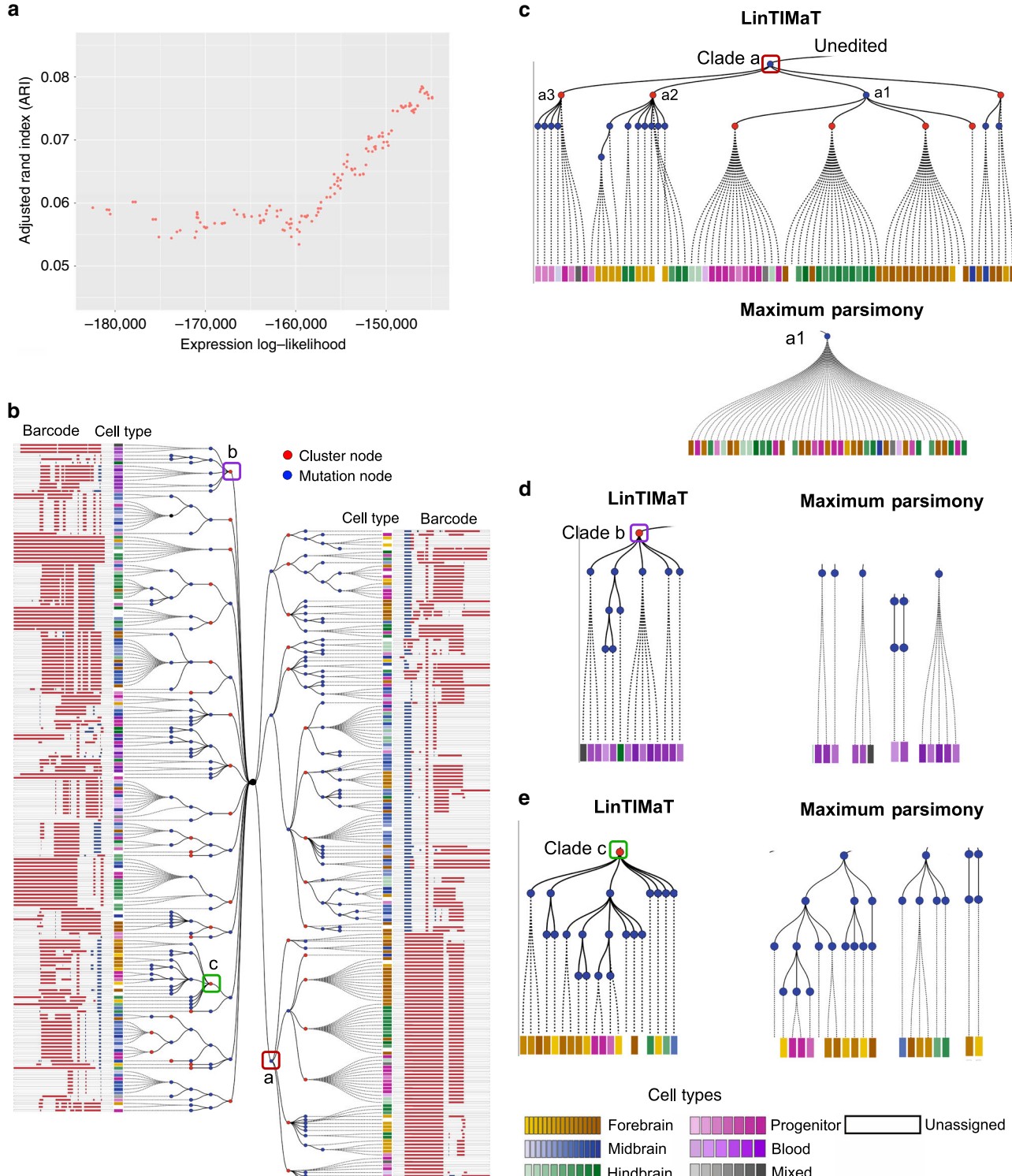

**Fig. 3 Reconstructed cell lineage for a single juvenile zebrafish brain (ZF3) from scGESTALT dataset. a** Adjusted Rand Index (ARI) which measures the agreement between cell types in the tree clusters and cell types assigned by the original paper[20] as a function of the likelihood computed by LinTIMaT. The fact that as the likelihood increases the ARI increases as well indicates that the target function of LinTIMaT is capturing biologically relevant relationships between cells. **b** Reconstructed cell lineage tree for ZF3 built on 376 cells. Blue nodes represent Cas9-editing events (mutations) and red nodes represent clusters inferred from transcriptomic data. Each leaf node is a cell, represented by a square, and its color represents its assigned cell type as indicated in the legend. The mutated barcode for each cell is displayed as a white bar with insertions (blue) and deletions (red). **c** By using transcriptomic data LinTIMaT is able to further refine subtrees in which all cells share the same barcode, which can help overcome saturation issues. **d**, **e** Example subtrees displaying LinTIMaT's ability to cluster cells with different barcodes together based on their cell types. In contrast, maximum parsimony puts these on distinct branches.

(progenitor, hindbrain, and forebrain) under the same mutational node. Similarly for ZF1, in the original MP lineage tree, clade a consisted of 198 cells including mostly forebrain and progenitor cells. LinTIMaT lineage tree successfully divided them into multiple subtrees, with the largest mainly containing forebrain neuron cells and the other subtrees mostly containing different types of progenitor cells (Supplementary Fig. 11a). In addition, LinTIMaT trees also contained examples where cells belonging to similar cell types but carrying different mutational barcodes are identified as a cluster instead of being placed on distant branches as done by MP. Clades b and c in ZF3 lineage tree (Fig. 3d, e) illustrate this scenario. In the LinTIMaT lineage tree for ZF3, clade b consists of mostly blood cells that carry different mutational barcodes. In MP lineage tree, these cells were placed in four distant branches, which did not convey the fact that they belong to the same cell type. However, LinTIMaT successfully grouped them together in a cluster of blood cells while preserving their mutational differences as illustrated by the mutation nodes being descendants of the cluster node. Similarly, for clade c most of the cells were forebrain neurons that were placed in three distinct branches in the MP lineage tree owing to their mutational differences. LinTIMaT successfully identified these cells as a cluster consisting of mostly forebrain neuron cells. Similar examples can be seen in the tree reconstructed by LinTIMaT for ZF1 (Supplementary Fig. 11b). We note that while the LinTIMaT reconstructed lineages displayed much better agreement with cell type coherence, this was not a function of ignoring mutational data. In fact, the trees inferred by LinTIMaT had higher likelihoods based on mutation alone (Supplementary Table 2) when compared to the trees reconstructed by MP[20]. In fact, for each fish, the MP lineage tree violated the Camin-Sokal parsimony criterion for some mutations that resulted in a low mutation log-likelihood. LinTIMaT lineage trees also revealed divergent lineage trajectories (Fig. 3b, c, Supplementary Discussion).

Following the analysis of [20], we also analyzed LinTIMaT and MP lineage trees for spatial enrichment of clusters by selecting groups of four or more cells. In both types of lineage trees, clusters were spatially enriched in hindbrain, forebrain and midbrain (Supplementary Fig. 12). However, LinTIMaT reconstructed lineages displayed better spatial enrichment. For ZF3, LinTIMaT lineage showed more enriched forebrain and hindbrain clusters compared to the barcode clusters in MP tree, whereas for ZF1, more hindbrain clusters were enriched in LinTIMaT lineage compared to the barcode clusters in MP tree. We also compared the lineages by assessing the functional significance of the clusters through Gene Ontology (GO) analysis. The clusters identified by LinTIMaT led to more significant enrichment of more GO functions compared to the barcode clusters in MP tree (Supplementary Fig. 13 and Supplementary Table 6).

To assess LinTIMaT's ability to generalize to other types of CRISPR-mutation data, we further applied it on data generated by the ScarTrace[22] method. Similar to scGESTALT, ScarTrace also uses CRISPR-Cas9 technology for introducing heritable mutations, though it uses a different lineage recording system where genomic sites located on in-tandem copies of a transgene are targeted for inserting mutations (also called scars). For this dataset, we applied LinTIMaT on two zebrafish samples, R2 and R3, for which the cells were sampled from adult brain and eyes. For each of these fishes, we selected 750 cells based on their cell types. The mutational dataset for R2 consisted of 133 unique barcodes and 78 unique scars, whereas that for R3 consisted of 85 unique barcodes and 50 unique scars. Applying LinTIMaT to this data resulted in highly branched multiclade lineage trees (can be visualized at https://jessica1338.github.io/LinTIMaT/). For

comparison, we also reconstructed lineage trees using MP for the two fishes. Similar to our observations for scGESTALT dataset, LinTIMaT was able to correctly separate different types of cells that were clustered together by MP as well as cluster cells that belonged to similar cell types but carried different mutational barcodes. We present a number of examples for these results in Supplementary Figs. 14–16. In addition, the lineage trees reconstructed by LinTIMaT for the ScarTrace datasets had higher likelihoods based on mutation alone (Supplementary Table 3) when compared to the trees reconstructed by MP. The much lower likelihoods of the MP trees were caused by the violation of Camin-Sokal parsimony criterion by multiple mutations (Supplementary Discussion).

**LinTIMaT successfully combines data from individual lineages.** Combining CRISPR-Cas9-mutation-based individual lineage trees is challenging since mutations are random and so differ for the same cell types between experiments. To address this, we used LinTIMaT to combine data from the replicates generated by scGESTALT and ScarTrace to infer invariant lineages for the development of juvenile zebrafish brain and the development of zebrafish brain and eyes respectively.

For the scGESTALT dataset, LinTIMaT inferred 113 clusters for ZF1 and 42 clusters for ZF3, out of which 33 clusters were found to be preserved in both lineages. Using these, LinTIMaT inferred an invariant lineage tree (Fig. 4) with 33 leaves each of which represents a matched pair of clusters from the individual fishes. We first evaluated the invariant lineage by computing its Adjusted Rand Index (ARI) based on the 63 cell types obtained by[20]. Our analysis showed that, despite the individual fishes having different spatial distribution of cells (e.g., ZF1 had more forebrain cells and ZF3 had more hindbrain cells), the ARI for the invariant lineage (0.079) was comparable to the individual LinTIMaT lineages (0.084 and 0.076 for ZF1, and ZF3, respectively) and higher than both individual MP lineages (0.061 and 0.056 for ZF1 and ZF3, respectively). While the invariant lineage preserved some of the ancestor-descendant relationship of the individual lineages (Supplementary Fig. 17), it also placed similar cell clusters from different branches of the individual trees under the same subtree (Supplementary Fig. 18). Thus, in addition to enabling the integration of data across experiments, by using more data, the invariant lineage tree method also improved the placement of the matched clusters on the individual trees themselves.

Spatial enrichment analysis revealed the matched clusters in the invariant lineage to be enriched in all three regions of brain (hindbrain, forebrain, and midbrain) as shown in Fig. 5a. The invariant lineage showed more enriched hindbrain clusters compared to that of ZF1 and more enriched forebrain clusters compared to that of ZF3.

To determine the biological significance of the clusters identified by the invariant lineage, we performed GO analysis (Methods) on matched clusters that contained more than 10 cells. We also filtered the matched clusters where the individual cluster contained fewer than three cells. We selected all GO terms related to the three major cell types (neuron, blood and progenitor) present in the data (see Supplementary Tables 4, 7, and 8 for the keywords and list of GO terms). Figure 5b illustrates the enrichment of the GO terms in the clusters in terms of p-values. The invariant clusters showed coherent enrichment of GO terms for all three major cell types. For example, clusters c23 (forebrain), c17 (midbrain and forebrain), and c2 (midbrain) had high p-value for the GO terms related to neuron but low p-value for GO terms related to blood and progenitor. Clusters c4 and c10 consisting mostly of forebrain neurons and some

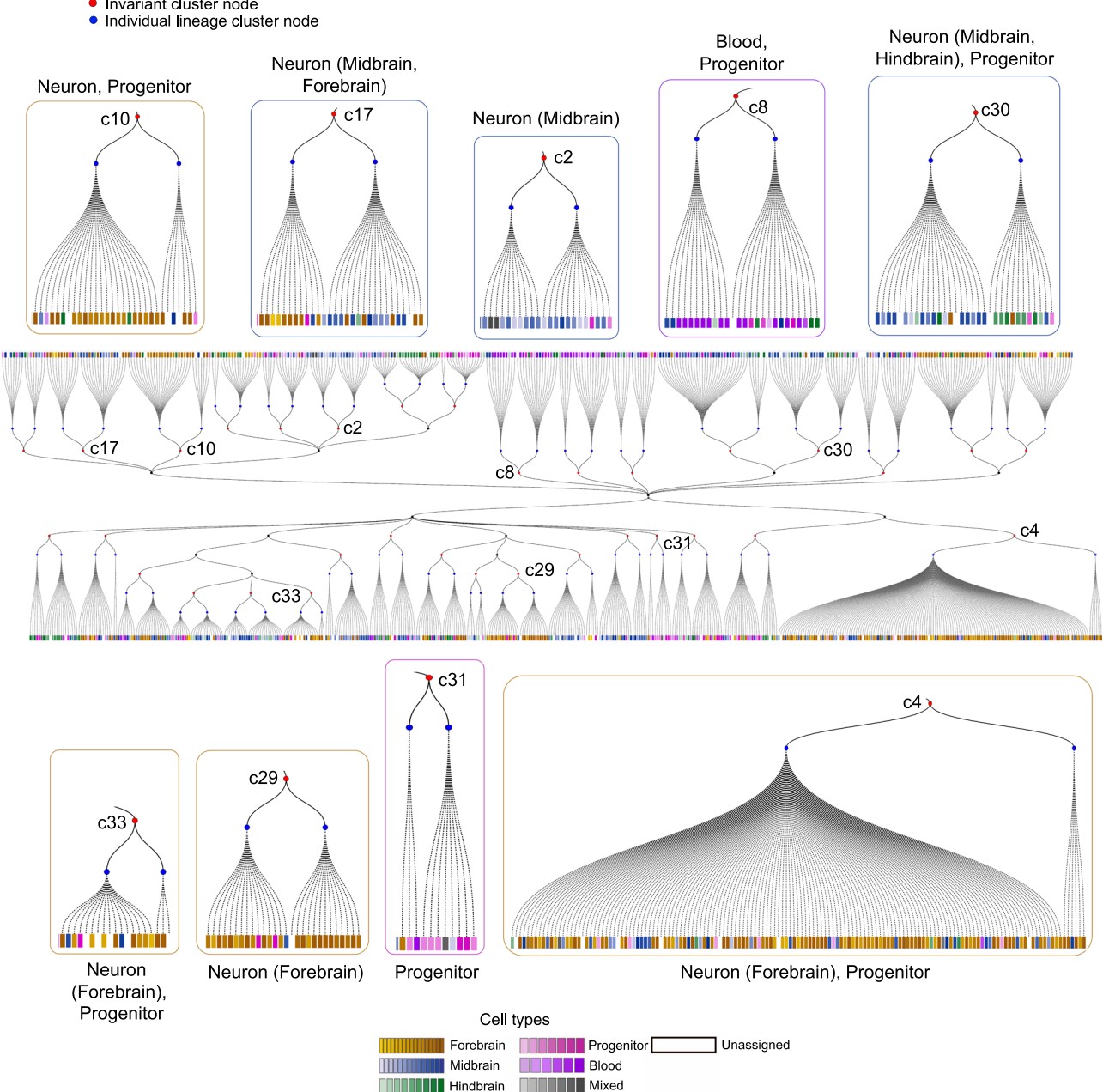

**Fig. 4 Invariant lineage tree for juvenile zebrafish brain for scGESTALT dataset.** The two-sided tree in the middle represents the invariant lineage tree generated by LinTIMaT by combining the individual trees for ZF1 and ZF3. Blue nodes here represent the clusters from individual fishes (left node: ZF1, right node: ZF3), and red nodes represent the matched invariant clusters. Each leaf node is a cell, represented by a square, and its color represents its cell type as indicated in the legend. Subtrees illustrate examples of invariant clusters preserved in the individual lineage trees.

progenitor cells showed enrichment of mostly neuron related GO terms and some progenitor GO terms. Similar GO enrichment was observed for cluster c30 that mostly consisted of midbrain and hindbrain neurons and some progenitor cells. The cluster c31 consisting of mostly progenitor cells displayed more enrichment of the progenitor GO terms. Clusters c8, and c32 that consisted mostly of blood and progenitor cells showed enrichment of GO terms related to these two cell types. The invariant clusters also uncovered additional GO functions that were not enriched in individual tree clusters (Supplementary Table 10). The coherence of enrichment can also be observed in the proportion of the GO terms related to the three major cell types (Supplementary Fig. 19). Clusters in the individual lineage trees also showed

enrichment of the three cell types. However, the invariant lineage clusters uncovered more GO terms with more significant *p*-values compared to the individual lineage clusters.

We further reconstructed an invariant lineage for the ScarTrace dataset. For this data, LinTIMaT inferred 83 clusters for R2 and 90 clusters for R3; and identified 52 matched clusters, which were used to reconstruct the invariant tree (visualized at https://jessica1338.github.io/LinTIMaT/) with 52 leaves, each of which represents a matched pair of clusters from the individual fishes. To determine if the invariant clusters inferred by LinTIMaT uncover functions coherent with the types of cells, we performed GO analysis by selecting all GO terms related to the three major cell types in the data (neuron, immune and eye, Supplementary

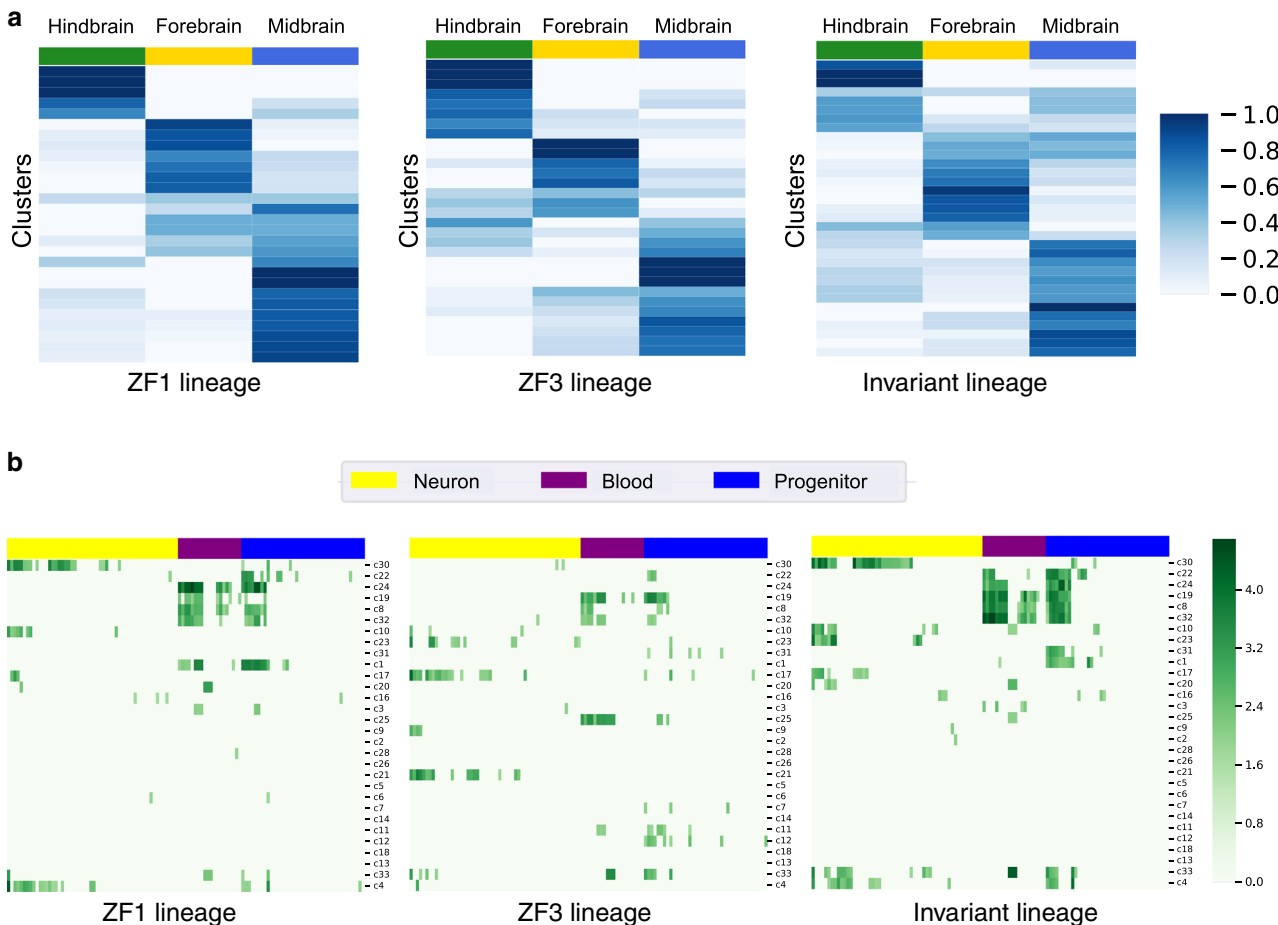

**Fig. 5 Functional analysis of cell clusters for scGESTALT datasets. a** Heat map of the distribution of cell clusters for each region of the brain (columns). Cell types were classified as belonging to the forebrain, midbrain or hindbrain, and the proportions of cells within each region were calculated for each cluster. Each row sums to 1. Region proportions were colored as shown in key. The leftmost panel shows the heat map for the clusters in ZF1 lineage (subsampled), middle panel shows the heat map for ZF3 lineage and the rightmost panel shows the heat map for the invariant lineage. **b** Heat map of the $p$-values ($\sqrt{-\log(p\,\text{value})}$, higher value means more significant) for GO terms for invariant clusters. Adjusted $p$-values for GO terms were obtained from g: Profiler[46]. *P*-values are calculated using the hypergeometric distribution. *P*-values are adjusted using the g:SCS algorithm[49]. Rows represent invariant clusters and columns represent different GO terms (Supplementary Table 8). Yellow, purple and blue columns correspond to GO terms related to neurons, blood and progenitors respectively. The leftmost panel shows the heat map for ZF1, middle panel for ZF3 and the rightmost panel for the invariant tree. As can be seen, the invariant tree correctly combines the unique terms identified for each tree. On one hand, it is able to identify neuron clusters, which are well represented in ZF3 but not in ZF1. On the other hand, it is able to identify progenitor clusters which are not well represented in ZF3.

Tables 5 and 9). As can be seen in Supplementary Fig. 20, the invariant clusters showed better enrichment of GO terms for all three major cell types. For example, in clusters c7 and c21, neuron related GO terms were enriched. Clusters c43, c19, and c9 showed enrichment of GO terms related to immune cell types. Clusters c47, c11 and c52 showed enrichment of GO terms related to eye cell types. The invariant clusters also uncovered additional GO terms that were not identified as significant when using the individual tree clusters (Supplementary Table 11). For example, two invariant clusters (c33 and c44) were found to be associated with erythrocyte and myeloid cell development (corrected $p$-value ≤ 0.027). Two other clusters (c7 and c31) were found to be associated with positive regulation of synaptic transmission and photoreceptor cell outer segment organization ($p$-values ≤ 0.004 and ≤0.0028, respectively). In both cases, cells related to these categories were not identified in the individual fish trees.

## Discussion
Recent studies[20–22] combine two complementary technologies, CRISPR-Cas9 genome editing and scRNA-seq for elucidating developmental lineages at whole organism level. These experimental techniques rely on introducing random heritable mutations during cell division using CRISPR-Cas9 and lineage trees are reconstructed based on these mutations using traditional phylogenetic algorithms[24].

While this exciting new direction to address a decades old problem in-vivo has already led to several interesting insights into organ development in multicellular organisms, it suffers from several challenges that make it difficult to accurately reconstruct lineages and to combine trees reconstructed from repeat experiments. First, the tree reconstruction is performed solely based on recovered mutation data, which might be noisy. In addition, the space for the mutations is limited resulting in saturation restricting the ability to track further subdivision of cells at later stages. Finally, due to the random nature of these mutations, it is impossible to utilize them to reconstruct a consensus lineage tree by combining data from repeated experiments of the same species, in contrast to most phylogenetic studies[33]. While very recently, computational methods that address some of these challenges are being developed[21,34,35], they still rely only on mutation information.

To address these issues, we developed a statistical method, LinTIMaT, which directly incorporates expression data along with mutation information for reconstructing both, individual and species-invariant lineage trees. Our method defines a global likelihood function that combines both mutation agreement and expression coherence.

We first used data from *C. elegans* for which ground truth lineage is known to validate the underlying assumption of our method: that expression coherence can indeed help in overcoming mutation data noise. As we show, for several possible noise factors that can appear in CRISPR-Cas9 lineage experiments, LinTIMaT was able to successfully improve the reconstruction of the lineage tree by using the additional expression information. We next used LinTIMaT on more complex data. While the ground truth for these lineages is unknown, we have shown that the trees reconstructed by LinTIMaT are as good as the best mutation-only lineage trees while they greatly improve over mutation-only lineages in terms of expression coherence, clade homogeneity and functional annotations. In addition, by employing agreement based on expression data, we could further reconstruct a species-invariant lineage that successfully retained the original tree branching and cell clusters common in each individual while improving on the individual lineages by uncovering more biologically significant GO annotations corresponding to different major cell types.

Our analysis shows that gene expression data can be very useful for selecting between several lineages with equivalent explanation of the mutation data. Since traditional phylogenetic maximum parsimony algorithms[24] as used in current studies[20] end up selecting a solution that is only slightly better or equivalent compared to several competing ones (though can be very different), the ability to use additional information (in our case gene expression) to select between these equally likely lineage trees is a major advantage of LinTIMaT. LinTIMaT's Bayesian hierarchical model for gene expression data also provides a statistical method for inferring cell clusters with coherent cell types from the lineage tree. While it is not clear yet if all organisms follow the same detailed developmental plan as *C. elegans*[36], the ability to combine lineage trees studied in multiple individuals of the same species can lead to more general trees that capture the major branching events for the species. In addition, invariant trees can be used to improve branchings in the individual trees by combining information from multiple experiments. To the best of our knowledge, LinTIMaT's solution, which is based on iteratively matching cell clusters based on their expression, is the first to enable the reconstruction of such invariant lineage trees from experiments that simultaneously profile lineage recordings and single-cell transcriptomes.

While LinTIMaT worked well on the datasets it was tested on, there exist potential situations for which our approach might face difficulties. It is currently unclear if cell trajectories inferred by transcriptional state and lineage should be concordant in all cases. As we showed, LinTIMaT can correctly identify lineage relationships even if such differences exist, but it is still possible that in some cases the use of expression data may lead to less accurate reconstructions. LinTIMaT uses a user defined parameter for weighting the contribution of the mutation and expression data. While our analysis indicates that the outcome is usually robust to changes of this parameter, a user can change the value of the weight parameter to reflect their belief about the importance of the two data types for their experiments. Another potential problem arises from our selection of clusters for reconstructing the invariant lineage tree. Since we only use clusters observed in all individual trees, the method may leave out several key clusters (or lineages) if their expression levels are not well conserved between different organisms from the same species.

The application of LinTIMaT to zebrafish brain development illustrates its potential in delineating lineage relationships in complex tissues. The method is general and, as we showed, can work with data for several different related technologies. While the joint profiling of lineage recordings and single-cell transcriptomes by experimental methods such as scGESTALT laid the foundation for generating data suitable for identifying cellular relationships during development and disease, LinTIMaT provides a unique computational approach for utilizing such data for accurate lineage reconstruction. As the usage of the experimental methods expands from zebrafish to other model organisms and human organoid samples[3], LinTIMaT would serve as a powerful component in the biologists' toolbox in reconstructing more accurate and detailed lineages for investigating normal as well as pathological development.

## Methods

**Processing of the input data.** LinTIMaT is designed for single-cell datasets in which both edited barcode and scRNA-seq data are available from the same cell. For scGESTALT datasets, each CRISPR-Cas9 mutation event (edit) has variable length and a single event could span across multiple adjacent sites. To construct a lineage tree from the mutation data we first count the number of unique synthetic markers (Cas9 edits) that occur in the 9 mutation sites. For each cell, the mutated barcode is represented by a binary vector of length equal to the number of unique synthetic markers, where each bit represents the state of a synthetic marker. For example, for ZF1 in the scGESTALT dataset there are 324 entries in this vector for each cell. Similarly, for ScarTrace dataset, we also use a binary vector with length equal to all unique mutations to represent the mutated barcode for each cell, and each bit of the binary vector represents whether or not the barcode contains the mutation event in at least one of its target sites. We use the mutation data to construct a paired-event matrix, $\mathcal{E}_{B \times S}$ for $B$ unique barcodes and $S$ unique editing events (synthetic markers), and an imputed gene-expression matrix, $\mathcal{Y}_{N \times G}$ for $N$ cells and $G$ genes.

Each row of the paired-event matrix $\mathcal{E}$, corresponds to a mutated barcode (or allele) and each column corresponds to a unique editing event. An entry $e_{bs}$ of $\mathcal{E}$ is a binary variable that denotes the presence or absence of marker $s$ in barcode $b$ (1 or 0). Each cell $c$ is associated with one, and only one, of the $B$ unique barcodes. As a result, each barcode represents a group of cells. For each cell $c = 1, \ldots, N$, $z_c$ denotes the barcode $b$ profiled for that cell, $z_c = b$, where $b \in \{1, \ldots, B\}$. Thus, the matrix $\mathcal{E}$ can be transformed to an $N \times S$ matrix for $N$ cells and $S$ markers, where the row $c$ will correspond to the barcode $z_c$ associated with cell $c$.

The other type of data our method uses is scRNA-seq data. In general, the method can work with any such data. For the specific datasets used in this paper, we observed a high dropout rate (94% entries were 0). To address this issue we tested a number of imputation methods (see "Supplementary Methods" and Supplementary Fig. 21) and selected DrImpute[37] for imputation. DrImpute first clusters the data, and then each zero expression value is imputed with the mean gene expression of the cells in the cluster the cell belongs to. Next, we normalized the expression of each cell and log2-transformed the results ("Supplementary Methods").

**Likelihood of a cell lineage tree.** As mentioned in the Introduction, our method aims to reconstruct a cell lineage tree by combining two complementary types of data. For this, we defined a joint likelihood function for the two data types and then search the space of possible trees for a model that maximizes the likelihood function. We first describe the likelihood function for each of the data types and then discuss how to perform a search for maximizing the joint likelihood to reconstruct the most likely tree.

**Cell lineage tree.** We assume that the cell lineage tree is a rooted directed tree $\mathcal{T}$. The root of this lineage tree denotes the initial cell that does not contain any marker (or editing event). The leaves of this tree denote cells profiled in the experiment. Cells go through the differentiation process along the branches of the lineage tree and as part of this process acquire the synthetic mutations (edits) induced by Cas9. Some of the internal nodes in the cell lineage tree represent the unique mutated barcodes shared by the leaves (cells) under that specific internal node. For ease of computation, we first reconstruct a rooted binary lineage tree and later eliminate the internal branchings that are not supported by any synthetic mutations.

**Mutation likelihood.** The first component of the likelihood function evaluates the likelihood of the cell lineage tree based on the mutation data. The mutations induced by Cas9 are irreversible since the Cas9 protein cannot bind to the target sites once changed. To account for this, we impose a Camin-Sokal parsimony criterion[38] on each synthetic mutation. This criterion states that each synthetic

mutation can be acquired at least once along the lineage but once acquired they are never lost. We also assume that the synthetic mutations are acquired independently and parsimoniously as higher number of mutations along the branches of the cell lineage indicates a more complex mutational history which is less likely. For a given cell lineage tree $\mathcal{T}$, we first use Fitch's algorithm[39] to assign ancestral states for each marker to each internal node of the tree satisfying maximum parsimony. Such an assignment, $\mathcal{A}$ results in the least number of mutations on the given tree. The mutation likelihood ($\mathcal{L}_{\mathrm{M}}$) of the cell lineage tree is then given by

$$\mathcal{L}_{\mathrm{M}}(\mathcal{T}) = P(\mathcal{E}|\mathcal{T}, \mathcal{A}) = \prod_{s=1}^{S} P(\mathcal{E}_{*s}|\mathcal{T}, \mathcal{A}_s) \qquad (1)$$

where $\mathcal{E}_{*s}$ is the observed data for marker $s$ which is a vector corresponding to $N$ values for $N$ cells. $\mathcal{A}_s$ denotes the parsimonious assignment of ancestral states for all internal nodes for marker $s$. For an internal node $v$ with children $u$ and $w$, $L_s^v(\mathcal{A})$ denotes the partial conditional likelihood for marker $s$ defined by

$$L_s^v(\mathcal{A}_s^v = x) = P(\mathcal{E}_s^v|\mathcal{T}, \mathcal{A}_s^v = x) \qquad (2)$$

where $\mathcal{E}_s^v$ denotes the restriction of observed data for marker $s$, $\mathcal{E}_{*s}$ to the descendants of node $v$ subject to the condition that $\mathcal{A}_s^v = x$ is the ancestral state for marker $s$ assigned by Fitch's algorithm, $x \in \{0, 1\}$. $L_s^v$ gives the likelihood for marker $s$ for the subtree rooted at node $v$, given the assignment of ancestral states by Fitch's algorithm.

The likelihood for the full observed data $\mathcal{E}_{*s}$ for marker $s$ is given by

$$P(\mathcal{E}_{*s}|\mathcal{T}, \mathcal{A}) = L_s^r(\mathcal{A}_s^r = 0) \qquad (3)$$

where $r$ is the root of the lineage tree. Since, the root of the tree does not contain any synthetic mutation, $\mathcal{A}_s^r = 0, \forall s \in \{1, 2, \dots, S\}$. For any internal node $v$ with children $u$ and $w$, the partial conditional likelihood satisfies the recursive relation

$$L_s^v = \left[ P_{t_{\mathcal{A}_s^v \to \mathcal{A}_s^u}} L_s^u \right] \left[ P_{t_{\mathcal{A}_s^v \to \mathcal{A}_s^w}} L_s^w \right] \qquad (4)$$

$P_{t_{\mathcal{A}_s^v \to \mathcal{A}_s^u}}$ and $P_{t_{\mathcal{A}_s^v \to \mathcal{A}_s^w}}$ denote the transition probabilities on branches that connect $v$ and $u$, and $v$ and $w$, respectively. For each synthetic mutation $s$, we define a transition probability matrix given by

$$P_t^s = \begin{bmatrix} 1 - m_s & m_s \\ 0 & 1 \end{bmatrix} \qquad (5)$$

where $m_s$ denotes the fraction of cells harboring $s$ and $P_t^s(i, j)$ denotes the probability of transition from state $i$ to state $j$ along any branch of the tree. If a mutation assignment violates the Camin-Sokal parsimony criterion (i.e. a mutation is reversed), the log-likelihood is heavily penalized ($-100000$) so that LinTIMaT prefers the tree without such violation.

For each leaf $l$ of the tree, the partial likelihood is set to $L_s^l = 1$. It is important to note that our mutation likelihood function does not explicitly model the editing rate at each CRISPR target.

If a mutation $s$ is affected by dropout in some cells, the mutation state appears to be unmutated in those cells and in effect, observed $m_s$ will be lower than the actual mutation fraction $a_s$ (true fraction of cells harboring $s$). For the datasets we analyzed, we did not have an estimate of the dropout rate. If for some dataset a good estimate of the dropout rate is available, it can be incorporated in the transition probability matrix by modeling $m_s$ as a function of $a_s$ and dropout rate where $a_s$ is modeled as a beta distributed random variable whose values are simultaneously estimated while learning the lineage.

**Expression likelihood**. For the expression data likelihood, we model the lineage as a Bayesian hierarchical clustering (BHC)[25] of the cells and used the likelihood formulation provided by BHC. BHC is a bottom-up agglomerative clustering method that iteratively merges clusters based on marginal likelihoods. Following several other methods we assume a diagonal matrix when computing gene expression variance for each internal and leaf node[40,41]. Following BHC algorithm, we compute the marginal likelihoods of all the partitions consistent with the given lineage tree based on a Dirichlet process mixture model. The expression likelihood ($\mathcal{L}_{\mathrm{E}}$) for the complete dataset is given by the marginal likelihood for the root of the tree and it essentially provides a lower bound on the marginal likelihood of a Dirichlet process mixture model.

$$\mathcal{L}_{\mathrm{E}}(\mathcal{T}) = P(\mathcal{Y}|\mathcal{T}) = \mathcal{L}_{\mathcal{G}}^r \qquad (6)$$

where $\mathcal{Y}$ is the $N \times G$ gene-expression matrix, $\mathcal{G}$ is the set of $G$ genes and $P(\mathcal{Y}|\mathcal{T})$ is the expression likelihood for the lineage tree which is also the marginal likelihood ($\mathcal{L}_{\mathcal{G}}^r$) for the root of the tree.

For an internal node $v$ with children $u$ and $w$, $\mathcal{T}^v$ denotes the subtree rooted at $v$. Let $\mathcal{Y}^v \subset \mathcal{Y}$ be the set of gene expression data at the leaves under the subtree $\mathcal{T}^v$ and $\mathcal{Y}^v = \mathcal{Y}^u \bigcup \mathcal{Y}^w$. To compute the marginal likelihood for node $v$ ($\mathcal{L}_{\mathcal{G}}^v$), we compute the probability of the data under two hypotheses of BHC. The first hypothesis, $\mathcal{H}_1^v$ assumes that each data point is independently generated from a mixture model and each cluster corresponds to a distribution component. This means that the data points $\mathbf{y}^{(i)}$ in the cluster $\mathcal{Y}^v$ are independently and identically generated from a probabilistic model $P(\mathbf{y}|\theta)$ with parameters $\theta$. Thus, the marginal

probability of the data $\mathcal{Y}^v$ under the hypothesis $\mathcal{H}_1^v$ is given by

$$P(\mathcal{Y}^v|\mathcal{H}_1^v) = \int P(\mathcal{Y}^v|\theta)P(\theta|\beta)d\theta$$

$$1em = \int \left[ \prod_{\mathbf{y}^{(i)} \in \mathcal{Y}^v} P(\mathbf{y}^{(i)}|\theta) \right] P(\theta|\beta)d\theta \qquad (7)$$

The integral in Eq. (7) can be made tractable by choosing a distribution with conjugate prior, as discussed in Supplementary Methods.

The alternative hypothesis $\mathcal{H}_2^v$ assumes that there are two or more clusters in $\mathcal{Y}^v$. Instead of summing over all (exponential) possible ways of dividing $\mathcal{Y}^v$ into two or more clusters, we follow the strategy in BHC[25] and sum over the clusterings that partition the data $\mathcal{Y}^v$ in a way that is consistent with the subtrees $\mathcal{T}^u$ and $\mathcal{T}^w$. This gives us the probability of the data under the alternative hypothesis

$$P(\mathcal{Y}^v|\mathcal{H}_2^v) = \mathcal{L}_{\mathcal{G}}^u \mathcal{L}_{\mathcal{G}}^w = P(\mathcal{Y}^u|\mathcal{T}^u)P(\mathcal{Y}^w|\mathcal{T}^w) \qquad (8)$$

In Eq. (8), $P(\mathcal{Y}^u|\mathcal{T}^u)$ and $P(\mathcal{Y}^w|\mathcal{T}^w)$ represent the marginal likelihoods of subtrees rooted at nodes $u$ and $w$ respectively. Combining the two likelihoods of the two hypotheses leads to a recursive definition of the marginal likelihood for the subtree $\mathcal{T}^v$ rooted at the node $v$

$$\mathcal{L}_{\mathcal{G}}^v = P(\mathcal{Y}^v|\mathcal{T}^v) = \pi_v P(\mathcal{Y}^v|\mathcal{H}_1^v) + (1 - \pi_v)P(\mathcal{Y}^u|\mathcal{T}^u)P(\mathcal{Y}^w|\mathcal{T}^w) \qquad (9)$$

where $\pi_v$ is a parameter for weighting the two alternatives and is defined recursively for every node. The recursive definition of $\pi_v$ for node $v$ is given by

$$\pi_v = \frac{\alpha\Gamma(n_v)}{d_v}, \quad d_v = \alpha\Gamma(n_v) + d_u d_w \qquad (10)$$

In Eq. (10), $\alpha$ denotes a hyperparameter, the concentration parameter of the Dirichlet process mixture model, $n_v$ is the number of data points under the subtree $\mathcal{T}^v$ and $\Gamma(.)$ is the Gamma function. For each leaf $l$, we set the values $\pi_l = 1$ and $d_l = \alpha$. Also, for each leaf $l$, the marginal likelihood ($\mathcal{L}_{\mathcal{G}}^l$) is calculated based on only the first hypothesis

$$\mathcal{L}_{\mathcal{G}}^l = P(\mathcal{Y}^l|\mathcal{H}_1^l). \qquad (11)$$

See Supplementary Methods for discussion on how the prior is set for this model.

**Combined likelihood**. For a given lineage tree, the joint log-likelihood ($\mathcal{L}_{\mathrm{T}}$) function for the mutation and expression data is a weighted sum given by

$$\mathcal{L}_{\mathrm{T}}(\mathcal{T}) = \omega_1 \log \mathcal{L}_{\mathrm{M}}(\mathcal{T}) + \omega_2 \log \mathcal{L}_{\mathrm{E}}(\mathcal{T}) \qquad (12)$$

The values of $\omega_1$ and $\omega_2$ are chosen so that the values of the two likelihood components stay in the same range. In our experiments, we have used $\omega_1 = 50$ and $\omega_2 = 1$ (see Supplementary Fig. 22). $\omega_1$ and $\omega_2$ are kept as user defined parameters whose values can be changed to reflect the prior belief about the importance of the two types of data used by LinTIMaT.

**Search algorithm for inferring lineage tree**. Searching for the optimal tree under a maximum-likelihood framework like ours is a NP hard problem[42]. We have thus developed a heuristic search algorithm which stochastically explores the space of lineage trees. The search algorithm consists of several stages as described below.

1. In the first step, we only focus on the barcodes and search for top scoring solutions. The search process starts from a random tree topology built on $B$ leaves corresponding to $B$ unique barcodes. In searching the barcode lineage tree, we employ the mutation likelihood function. In each iteration, a new barcode lineage tree, $\mathcal{T}_B'$ is proposed from the current tree $\mathcal{T}_B$ as we discuss below. If the proposed tree results in a higher likelihood, it is accepted, otherwise rejected. Instead of storing a single solution, we keep several of top scoring barcode lineage trees.

$$\mathcal{T}_B^{[1]}, \mathcal{T}_B^{[2]}, \dots, \mathcal{T}_B^{[t]} = \operatorname*{argmax}_{\mathcal{T}_B} \mathcal{L}_{\mathrm{M}}(\mathcal{T}_B) = \operatorname*{argmax}_{\mathcal{T}_B} P(\mathcal{E}_{B \times S}|\mathcal{T}_B, \mathcal{A}) \qquad (13)$$

2. Next, we utilize the expression data. As mentioned above, a barcode can be shared between multiple cells. We thus next search for the best cellular subtree ($\mathcal{T}_b$) for the set of cells associated with each mutated barcode $b$. We employ hill-climbing to obtain single solution for each barcode that harbors more than 2 cells.

$$\mathcal{T}_b = \operatorname*{argmax}_{\mathcal{T}} P(\mathcal{Y}_{(c|z_c=b)*}|\mathcal{T}) 2mm \forall b \in \{1, \dots, B\} \qquad (14)$$

3. In the third step, we construct complete cell lineage trees by attaching cellular subtrees for each barcode to barcode lineage trees. To obtain the cell lineage tree $\mathcal{T}_i$ from a barcode lineage tree $\mathcal{T}_B^{[i]}$, for each barcode $b$ harboring more than two cells, we choose the cellular subtree $\mathcal{T}_b$ inferred in step 2 and connect its root to the leaf in $\mathcal{T}_B^{[i]}$ that corresponds to $b$. For a barcode $b$ shared by two cells, the cells are connected to the leaf representing $b$ in $\mathcal{T}_B^{[i]}$ as children. This gives us $t$ full binary cell lineage trees corresponding to $t$ barcode lineage trees. Next, we evaluate the total log-likelihood of each of these cell lineage trees and choose the

best one.

$$\mathcal{T}^+ = \underset{\mathcal{T}_i, i=1,\ldots,t}{\mathrm{argmax}}\ \mathcal{L}_T(\mathcal{T}_i) \tag{15}$$

We also record the best mutation log-likelihood, $\mathcal{L}_M^{best}$ for the best cell lineage tree and define a threshold value for mutation log-likelihood

$$\mathcal{L}_M^{thr} = \mathcal{L}_M^{best} + thr \times \mathcal{L}_M^{best} \tag{16}$$

where, thr is a user-defined value close to 0.

4. In the final step, we perform another hill-climbing search to optimize the cell lineage tree $\mathcal{T}^+$ inferred in step 3 in terms of the joint likelihood function. The search starts from $\mathcal{T}^+$ and in each iteration, we propose a new cell lineage tree $\mathcal{T}'$ from the current tree $\mathcal{T}$ as we discuss below. For the new tree, we first ensure that the mutation log-likelihood of the new tree does not go below $\mathcal{L}_M^{thr}$. If this condition is satisfied and the total likelihood is improved then the new lineage tree is accepted. We stop the search if the total likelihood does not improve for a large number of iterations and return the best lineage tree achieved so far.

$$\mathcal{T}_{best} = \underset{\mathcal{T}}{\mathrm{argmax}}\ \mathcal{L}_T(\mathcal{T}) \tag{17}$$

**Tree search**. To explore the space of lineage trees, LinTIMaT employ two different types of moves that can make small and big changes in the tree topology. For this, we adopt two of the tree proposals described in ref. [43] for efficient exploration of tree space for Bayesian phylogenetic inference. Both of these moves are branch-rearrangement proposals that alter the topology of the lineage tree.

The first tree proposal is a swapping move called Stochastic Nearest Neighbor Interchange (stNNI). In this move, we choose an internal branch as the focal branch and stochastically swap the subtrees attached to the focal branch. This type of move results in minimal topology change and is used only in the second step of our algorithm that infers cellular subtree for each mutated barcode.

The second tree proposal is a pruning-regrafting move, namely Random Subtree Pruning and Regrafting (rSPR). In this move, we first randomly select an interior branch, prune a subtree attached to that branch, and then reattach the subtree to another regrafting branch present in the other subtree. The regrafting branch is also chosen randomly. This type of move can introduce a larger amount of topology change in the tree and this is used in step 1 and 4 of our search algorithm.

**Inferring clusters from cell lineage tree**. To obtain cell clusters from the inferred lineage tree, we employ the statistical model comparison criterion provided by the BHC model for gene expression data. For an internal node $v$ with children $u$ and $w$, we compute the probability of the data under two hypotheses. The first hypothesis suggests that all the cells under the node $v$ belongs to a single cluster. We compute the posterior probability ($r_v$) of this hypothesis using Bayes rule:

$$r_v = P(\mathcal{H}_1^v | \mathcal{Y}^v) = \frac{\pi_v P(\mathcal{Y}^v | \mathcal{H}_1^v)}{\pi_v P(\mathcal{Y}^v | \mathcal{H}_1^v) + (1 - \pi_v) P(\mathcal{Y}^u | \mathcal{T}^u) P(\mathcal{Y}^w | \mathcal{T}^w)} \tag{18}$$

The lineage tree can be cut at the nodes where $r_v$ goes from $r_v < 0.5$ to $r_v > 0.5$ to obtain clustering of cells.

**Combining lineage trees from multiple individuals to reconstruct an invariant lineage tree**. As mentioned in the Introduction, a key challenge when working with CRISPR mutation data is the fact that these are not the same across different experiments. Thus, standard phylogenetic invariant tree building cannot be applied to this data. Instead, given a set of lineage trees, $\{\mathcal{T}_1, \ldots, \mathcal{T}_I\}$ for $I$ individuals, we construct a single lineage tree $\mathcal{T}_{inv}$ that jointly explains the differentiation of these individual organisms. Individual lineage trees that are input to the invariant lineage reconstruction method are built on a leaf set of different number of cells. $\mathcal{T}_{inv}$ is constructed by following the steps below.

1. For each individual lineage tree $\mathcal{T}_i$, we infer the cell clusters based on gene expression data.
2. We remove all clusters that contain number of cells fewer than a pre-determined number, tc (we use tc = 3 in our analyses).
3. Next, we compute gene expression distances for cluster groups denoted as a tuple ($cl_1, cl_2, \ldots, cl_I$), where $cl_i$ is one of the remaining clusters from lineage $\mathcal{T}_i$ ($i \in \{1, 2, \ldots, I\}$). The gene expression distance for a cluster group is computed by summing the pairwise gene expression distances for all possible cluster pairs in the cluster group. The top $x$% cluster groups with the smallest distance are selected as the set of candidate preserved cluster groups ($x = 1$ was used in our analyses).
4. Using a greedy algorithm, we first select a set of cluster groups (based on gene expression distance) to be incorporated in the invariant lineage tree. The process of selecting the cluster groups is called greedy cluster matching and the clusters which form a selected cluster group are called matched clusters. All the candidate cluster groups are ranked in an ascending order

based on the expression distance. In the greedy matching, we first select the cluster group with the smallest distance. The algorithm continues to select the next ranked cluster group if none of its constituent clusters have been matched before. This process goes on until no more cluster groups can be selected (i.e. the constituent clusters of the remaining cluster groups have already been matched before). This process results in $K$ cluster groups (consisting of matched clusters from individual lineages) which are used in the invariant lineage tree.
5. For each individual lineage tree $\mathcal{T}_i$, we obtain the backbone tree $\mathcal{T}_i^c$ built using these $K$ clusters.
6. $\mathcal{T}_{inv}$ is a lineage tree built on a leaf set of $K$ clusters. We define a cluster matching $\mathcal{M}$ as a matching where each cluster (represented by leaf) in each individual lineage tree $\mathcal{T}_i^c$ is matched with a leaf of $\mathcal{T}_{inv}$. We reconstruct $\mathcal{T}_{inv}$ and a cluster matching $\mathcal{M}_{inv}$ by minimizing an objective function given by

$$\mathcal{T}_{inv}, \mathcal{M}_{inv} = \underset{\mathcal{T}^*, \mathcal{M}^*}{\mathrm{argmin}}\ \omega_1 \sum_{i=1}^{I} \mathcal{S}(\mathcal{T}^*, \mathcal{T}_i^c) + \omega_2 \sum_{j=1}^{K} \mathscr{E}(c_j) \tag{19}$$

where $\mathcal{T}^*$ is a candidate invariant lineage, $\mathcal{M}^*$ is a candidate cluster matching, $\mathcal{S}(\mathcal{T}^*, \mathcal{T}_i^c)$ denotes the sum of pairwise leaf shortest path distance between candidate invariant lineage $\mathcal{T}^*$ and individual lineage $\mathcal{T}_i^c$, $\mathscr{E}(c_j)$ denotes the sum of pairwise distance between the clusters of the individual lineage trees that match with cluster (or leaf) $c_j$ in the candidate invariant lineage. The objective function for searching the invariant lineage and the optimal cluster matching is described below in detail. We employ a two-step heuristic search algorithm for optimizing the objective function (described below).

**Objective function for optimizing invariant lineage tree**. The objective function for reconstructing the invariant lineage attempts to balance two competing issues. The first is that the invariant tree should be able to capture the branchings that are similar in each of the individual lineages. The second is that the agreement (in terms of expression) between nearby subtrees in the invariant tree would be high. Thus attempt to minimize two different distance functions to select the optimal tree. $\mathcal{D}_\mathcal{S} = \sum_{i=1}^{I} \mathcal{S}(\mathcal{T}^*, \mathcal{T}_i^c)$ computes the distance (or disagreement) between the topology of the invariant lineage and the backbone trees $\mathcal{T}_i^c$ obtained from the individual lineage trees. $\mathcal{D}_\mathcal{E} = \sum_{j=1}^{K} \mathscr{E}(c_j)$ is the other distance function which attempts to minimize disagreement between the gene expression values of matched clusters.

For computing $\mathcal{D}_\mathcal{S}$, we employ the sum of pairwise leaf shortest-path distance[44,45] between two trees as a distance measure for comparing two tree topologies. The shortest path distance $\delta_{ij}(.)$ between two leaves $c_i$ and $c_j$ in a tree is given by the sum of the number of edges that separate them from their most recent common ancestor. Overall pairwise leaf shortest-path distance between two trees is obtained by summing up the absolute differences between the shortest-path distances of all unordered pairs of leaves in the two trees

$$\mathcal{S}(\mathcal{T}_1, \mathcal{T}_2) = \sum_{i=0}^{K-1} \sum_{j=i+1}^{K} |\delta_{ij}(\mathcal{T}_1) - \delta_{ij}(\mathcal{T}_2)| \tag{20}$$

For computing $\mathcal{D}_\mathcal{E}$, we sum the pairwise distance between the clusters of the individual lineage trees that match with a leaf of the invariant lineage. $\mathscr{E}(c)$ is given by

$$\mathscr{E}(c) = \sum_{i=1}^{I-1} \sum_{k=i+1}^{I} e(l_i^c, l_k^c) \tag{21}$$

where $l_i^c$ and $l_k^c$ denote clusters in individual lineages that match with leaf $c$ in candidate invariant lineage. $e(.)$ denotes the Euclidean distance between the gene expression value of two clusters.

**Search algorithm for inferring invariant lineage**. We use a two-step heuristic search algorithm for inferring the invariant lineage and the corresponding cluster matching.

1. The first step employs an iterative search. In each iteration, we first find a better cluster matching (see "Supplementary Methods" for details) than the current matching $\mathcal{M}^*$, and then keeping this matching fixed, we improve the topology of the invariant tree. It is important to note that, a new cluster matching modifies both $\mathcal{D}_\mathcal{E}$ and $\mathcal{D}_\mathcal{S}$, whereas a new tree topology modifies only $\mathcal{D}_\mathcal{S}$. This iterative search goes on until cluster matching can not be improved further. Let us assume, $\mathcal{D}_\mathcal{E}^{best}$ is the distance corresponding to the best cluster matching achieved. We define a threshold value for the cluster matching distance

$$\mathcal{D}_\mathcal{E}^{thr} = \mathcal{D}_\mathcal{E}^{best} + thr \times \mathcal{D}_\mathcal{E}^{best} \tag{22}$$

2. In the second step, we try to improve the invariant lineage by improving the objective function $\mathcal{D}_\mathcal{S} + \mathcal{D}_\mathcal{E}$ using a stochastic search. In the joint ($\mathcal{T}^*, \mathcal{M}^*$) space, we consider two types of moves to propose a new configuration. In

each iteration, from the current configuration ($\mathcal{T}^*, \mathcal{M}^*$), we either propose a new matching (Supplementary Methods) $\mathcal{M}^*_{\text{new}}$ or a new tree topology $\mathcal{T}^*_{\text{new}}$ using the tree search moves. When a new matching $\mathcal{M}^*_{\text{new}}$ is proposed, we first ensure that the cluster matching distance for the new matching does not lead to values above the threshold $\mathcal{D}^{\text{thr}}_{\mathcal{E}}$. If this condition is satisfied and the objective function is minimized then the new matching is accepted. If the proposed tree topology $\mathcal{T}^*_{\text{new}}$ achieves lower value for the objective function, it is accepted. The search procedure terminates when the objective function does not improve or the maximum number of iterations has been reached.

**GO analysis on clusters identified by LinTIMaT.** To perform GO analysis on invariant lineage clusters, we first identify a set of differentially expressed (DE) genes based on $t$-test of two groups of cells. The first group consists of the cells in the invariant cluster and the second group includes all other cells in the dataset. From the set of DE genes, we further select the genes that have higher mean expression in the first group, with a $p$-value smaller than 0.05 (or top 500 if more than 500 genes achieve this $p$-value). Finally, we use gprofiler[46] to perform GO query for the genes selected for each cluster.

**Performance metrics for lineage reconstruction.** To assess the performance of a method in reconstructing the cell lineage, we employ two different metrics. First, we measure the accuracy of lineage reconstruction based on a metric used in ref. [28], which calculates the fraction of the non-trivial bipartitions in the ground truth lineage tree that are precisely recovered in the inferred lineage tree. In addition, we also compute the RF distance[31] between the true lineage tree and the inferred lineage tree for all methods. RF distance calculates the number of non-trivial bipartitions that differ between the inferred and true lineage trees (we normalize this using the total number of bipartitions in the two trees). For the binary lineage trees inferred by LinTIMaT, we compute RF distance (same as FP and FN distance). In contrast, since the lineage trees inferred by MP or NJ can potentially be nonbinary (when a complete lineage barcode is shared by more than two cells), we separately compute the FP and FN distances between the true lineage tree and the lineage inferred by MP or NJ.

**Analyzing the cell clustering performance of a lineage tree.** For assessing the cell clustering performance of a lineage tree, we use 63 cell types obtained by ref. [20] as ground truth and use Adjusted Rand Index (ARI) as the clustering metric following[47]. Basically, ARI is calculated based on the number of agreements and number of disagreements of two groupings, with randomness taken into account. ARI is defined as follows. Let $X = \{X_1, X_2, \ldots, X_r\}$, $Y = \{Y_1, Y_2, \ldots, Y_s\}$ be two groupings, where $X$ has $r$ clusters and $Y$ has $s$ clusters. We can set the overlap between $X$ and $Y$ using a table $N$ with size $r*s$, where $N_{ij} = |X_i \cap Y_j|$ denotes the number of objects that are common to both $X_i$ and $Y_j$. Let $a_i = \sum_j N_{ij}$, $b_j = \sum_i N_{ij}$, $n$ be the total number of samples, then ARI is given by

$$\text{ARI} = \frac{\text{Index} - \text{ExpectedIndex}}{\text{MaxIndex} - \text{ExpectedIndex}} = \frac{\sum_{ij}\binom{N_{ij}}{2} - (\sum_i \binom{a_i}{2} \sum_j \binom{b_j}{2})/\binom{n}{2}}{\frac{1}{2}(\sum_i \binom{a_i}{2} + \sum_j \binom{b_j}{2}) - (\sum_i \binom{a_i}{2} \sum_j \binom{b_j}{2})/\binom{n}{2}} \quad (23)$$

**Simulation of induced CRISPR-Cas9 mutations.** We simulate CRISPR mutations based on the 16 cell *C. elegans* lineage using a similar strategy outlined in[28]. For the simulation of CRISPR lineage recorders, each cell is represented as a vector of $m = 5$ target sites. The 16-cell lineage corresponds to a series of 4 cell divisions. The nonleaf nodes of the lineage represent the cells that underwent cell division. The root of the lineage represent the fertilized egg for which each CRISPR target is in an unmutated state. The branches that connect a nonleaf node to its children represent the branches where each unmutated target can mutate with a given probability $\mu$ denoting the mutation rate. Each target site can mutate to one of several possible mutated states. For each target site, the possible number of mutational events is chosen to be 8 and the different mutational events are considered to be equiprobable. After a mutation occurs at a target, it can no longer change in the absence of dropout. The simulation of CRISPR mutations starts from the root and follows the nonleaf nodes in the order of the cell division they represent (cell division at level 1 followed by division at level 2 and so on).

For simulating CRISPR mutations with varying mutation rate for different target sites, we first decide the value of mean mutation rate and standard deviation. Based on these two, we define a Beta distribution from which the mutation rate for each target is sampled.

To introduce mutation dropouts, we first define dropout rate as the ratio of the expected number of dropout events and the number of internal branches in the cell lineage. Dropouts are introduced in the lineage with probability equal to dropout rate and can only affect the target sites that have been already mutated. Whenever dropout happens at a target site, its previous lineage recording gets erased.

**Reporting summary.** Further information on research design is available in the Nature Research Reporting Summary linked to this article.

## Data availability

The high-throughput datasets used in this study were previously deposited in the Gene Expression Omnibus under accession numbers GSE77944, GSE105010, and GSE102990. Lineage trees are available for exploring at https://jessica1338.github.io/LinTIMaT/.

## Code availability

LinTIMaT has been implemented in Java and is freely available at https://github.com/jessica1338/LinTIMaT, under the MIT license. This implementation uses the PhyloNet[48] library and the Apache Commons Math package (https://commons.apache.org/proper/commons-math/).

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

## Acknowledgements

This work was partially funded by the National Institutes of Health (NIH) [grants 1R01GM122096 and OT2OD026682 to Z.B.-J.].

## Author contributions

H.Z., C.L., and Z.B.-J. designed the study. H.Z., C.L. and Z.B.-J. developed the model and algorithm. H.Z. and C.L. implemented the software and performed all experiments. All authors wrote and approved the paper. The order of authorship of H.Z. and C.L. was determined by a coin flip.

## Competing interests

The authors declare no competing interests.
