## [Peer Review File · Nature Communications]

Reviewers' comments:

Reviewer #1 (Remarks to the Author):

Zafar and Lin present an improved reconstruction approach for CRISPR/Cas9 lineage tracing method LimTIMaT. Their method constructs trees based on CRISPR edits and then refines the tree using expression data. The paper is comprehensive and details in many places where the LimTIMaT method improves tree reconstruction over maximum parsimony methods. LimTIMaT and the concept of refining lineage with the transcriptome is clearly an important step forward for the field of CRISPR/Cas9 lineage tracing. Overall the paper is well-written but the language could be more formal in some places (highlighted below). There are also a number of places where favorable outcomes are highlighted and a more rounded picture should be painted in the text. After addressing the review points below, and those highlighted by other reviewers, I believe this paper should be published.

-Aaron McKenna

Major points:

Why does the method do worse than normal maximum parsimony at some mutations rates? For instance in figure 2b, at 0.20, MP seems to outperform LimTIMaT. This seems a bit weird given the extra information LimTIMaT has available. It's important to explain why, and not gloss over this point in the text (maybe there is a future direction to highlight from this).

You show the effect of the dropout rate on both maximum parsimony and LimTIMaT at an error rate of 0.15, the most advantageous error rate for LimTIMaT. Could you include a supplemental figure with this plotted over the same error rates in figure 2b?

Some of the code to draw trees in your repository is taken verbatim (as full files) from our previous papers. We certainly should have attached a license (our fault), but you should indicate the source of this code in the codebase.

The figure legends should be more informative in some places. For instance figure 3: "(A) Adjusted Rand Index (ARI) as a function of the expression likelihood score calculated by LinTIMaT. The fact that both improve suggests a good agreement between the resulting tree and cell types." It would be more informative as something like "(A) The adjusted Rand index (ARI) calculated for cell-type similarity within subtree clusters. The ARI increases with improved LimTIMaT likelihood values." (not verbatim, just something like that). It's nice when the reader can understand the plot just from the legend and some improvements to the legend text would help.

Figure 4 is pretty hard to interpret. I understand we included it in our paper which makes for a nice comparison, but it's really hard to extract meaning from this. I might recommend moving this to the supplement or providing some aggregate measure of the improvement. Otherwise, it just leaves the reader guessing at the message here, as the text only provides some chosen examples of what improved. This goes for figure 6a as well, though that figure is probably easier to interpret, though I'm still stuck looking at the consensus tree and wondering why it seems more muddled than the two individual fish. The text offers an explanation that could be more explicit for this plot (top of page 11).

You say that no methods have been developed to meet the challenges of CRISPR/Cas9 barcoding (paragraph 2 in the discussion), but there are at least two I can think of. Reference 21 and Feng et. al 2019 on Biorxiv both propose solutions. Feng isn't peer-reviewed, but (1) both should be mentioned, and (2) can you do a quick comparison against the method in reference 21? You can ignore this if it's too much work but might be a nice highlight of the power of your method. Maybe it is also worth mentioning that maximum parsimony methods have assumptions, and you have to be aware of in your paper (the comparison in ref. 21 was flawed for this reason).

It would be worth mentioning in the discussion that the approach of refining a lineage with cell transcriptomes has a potential downfall. It is currently unclear if cell trajectories inferred by transcriptional state and lineage should be concordant in all cases, and although this method likely improves trees on the whole, it may obscure places where the cell lineages have divergent cell types late in development.

Minor points:

Line numbers are very helpful in review (for next time)

There is some use of imprecise language throughout the paper. For instance "LinTIMaT achieved better accuracy in lineage reconstruction compared to that of MP (for 0.2 they are almost the same, though for lower rates LinTIMaT is significantly better)." It's fine to say that LinTIMaT scored worse here, and again to the major point, worth explaining. The language of 'significant' and 'almost the same' should be cleaned up, as people would expect significant in this context to be statistical significance.

When you analyzed the CRISPR barcodes with variable editing rates (midway through page 6), does LinTIMaT provide an estimate of the editing rate at each target? Was it accurate?

Include references to the figures for all the examples. For instance, when you talk about her4.1+ positive (figure 3, not mentioned), it took me a minute to figure out what figure you meant, as the text just walked through figure 4.

Point-by-Point Responses for Reviewers

NCOMMS-19-29009

Single-cell Lineage Tracing by Integrating CRISPR-Cas9 Mutations with Transcriptomic Data

Hamim Zafar; Chieh Lin; Ziv Bar-Joseph

Dear Editors,

We thank the reviewers for their time and effort in providing us with valuable feedback for improving our manuscript. We have fully considered their comments and did the following to address their comments

1. We performed several additional simulation experiments (“convergence” and “divergence” in lineage as suggested by reviewer #2) to evaluate LinTIMaT’s ability to infer divergent lineage of similar celltype and convergent lineage of different cell types (scenarios that violate LinTIMaT’s model assumptions on gene-expression data).
2. We have clearly defined the notion of the lineage (previously called “consensus lineage”) reconstructed by combining multiple replicates of the same species as the lineage that captures a core “invariant” topology that is most probable in all cases. In the revised manuscript, we use the term “invariant lineage” to denote this. In order to achieve this, we have modified our previous consensus lineage reconstruction algorithm and updated the results accordingly.
3. We have further compared LinTIMaT’s performance against that of neighbor-joining algorithm on the *C. elegans* benchmark datasets.
4. We have performed the additional simulation experiments as suggested by reviewer #1.

While preparing the revised manuscript, we have updated the Main manuscript, Methods, Figures and Supplementary material by adding the results of our new analyses. The updated results and newly added paragraphs are written in blue font for the convenience of the reviewers. Below you will find our detailed responses to the reviewer’s comments and the results of the new analyses.

Reviewer #1

Remarks to the Author: Zafar and Lin present an improved reconstruction approach for CRISPR/Cas9 lineage tracing method LimTIMaT. Their method constructs trees based on CRISPR edits and then refines the tree using expression data. The paper is comprehensive and details in many places where the LimTIMaT method improves tree reconstruction over maximum parsimony methods. LimTIMaT and the concept of refining lineage with the transcriptome is clearly an important step forward for the field of CRISPR/Cas9 lineage tracing. Overall the paper is well-written but the language could be more formal in some places (highlighted below). There are also a number of places where favorable outcomes are highlighted and a more rounded picture should be painted in the text. After addressing the review points below, and those highlighted by other reviewers, I believe this paper should be published.

-Aaron McKenna

Response: We sincerely thank the reviewer for his positive remarks. We also thank him for providing valuable comments to improve our manuscript.

Major points:

R1.C1

Why does the method do worse than normal maximum parsimony at some mutations rates? For instance in figure 2b, at 0.20, MP seems to outperform LimTIMaT. This seems a bit weird given the extra

information LinTIMaT has available. It's important to explain why, and not gloss over this point in the text (maybe there is a future direction to highlight from this).

Response: We agree with the comment and based on it reanalyzed the dataset to test and compare the methods. Since it's a probabilistic algorithm we find that the results of LinTIMaT can slightly change based on the number of restarts (initializations) used by the method. Previously, we used only 3 restarts (with different seeds). When we increased the number of restarts, we found that LinTIMaT indeed performs better than maximum parsimony even for mutation rate = 0.2. Please see Fig. 2 and Supplementary Fig. S2 for details.

R1.C2

You show the effect of the dropout rate on both maximum parsimony and LinTIMaT at an error rate of 0.15, the most advantageous error rate for LinTIMaT. Could you include a supplemental figure with this plotted over the same error rates in figure 2b?

Response: As suggested, we performed additional dropout experiments where we vary mutation rate over the same values as in Figure 2b. Please see Supplementary Fig. S5 for details.

R1.C3

Some of the code to draw trees in your repository is taken verbatim (as full files) from our previous papers. We certainly should have attached a license (our fault), but you should indicate the source of this code in the codebase.

Response: We apologize for not indicating the source for our visualization code in our repository. We have clearly mentioned the source in our repository.

R1.C4

The figure legends should be more informative in some places. For instance figure 3: "(A) Adjusted Rand Index (ARI) as a function of the expression likelihood score calculated by LinTIMaT. The fact that both improve suggests a good agreement between the resulting tree and cell types." It would be more informative as something like "(A) The adjusted Rand index (ARI) calculated for cell-type similarity within subtree clusters. The ARI increases with improved LinTIMaT likelihood values." (not verbatim, just something like that). It's nice when the reader can understand the plot just from the legend and some improvements to the legend text would help.

Response: As suggested, we changed the legend for 3A. It now reads: "(A) Adjusted Rand Index (ARI) which measures the agreement between cell types in the tree clusters and cell types assigned by the original paper [Raj et al. 2018] as a function of the likelihood computed by LinTIMaT. The fact that as the likelihood increases the ARI increases as well indicate that the target function of LinTIMaT is capturing biological relevant relationships between cells."

R1.C5

Figure 4 is pretty hard to interpret. I understand we included it in our paper which makes for a nice comparison, but it's really hard to extract meaning from this. I might recommend moving this to the supplement or providing some aggregate measure of the improvement. Otherwise, it just leaves the reader guessing at the message here, as the text only provides some chosen examples of what improved. This goes for figure 6a as well, though that figure is probably easier to interpret, though I'm still stuck looking at the consensus tree and wondering why it seems more muddled than the two individual fish. The text offers an explanation that could be more explicit for this plot (top of page 11).

Response: As suggested, we have moved Fig. 4 to supplement. As for the consensus tree figure, based on related comments from Reviewer 2, we have changed the definition of the consensus tree to 'invariant lineage tree' and now make it clear that the tree only captures the invariant cell lineage branching and

clustering between multiple replicates of the same species. This is now mentioned both in the text and in the legend.

R1.C6

You say that no methods have been developed to meet the challenges of CRISPR/Cas9 barcoding (paragraph 2 in the discussion), but there are at least two I can think of. Reference 21 and Feng et. al 2019 on Biorxiv both propose solutions. Feng isn't peer-reviewed, but (1) both should be mentioned, and (2) can you do a quick comparison against the method in reference 21? You can ignore this if it's too much work but might be a nice highlight of the power of your method. Maybe it is also worth mentioning that maximum parsimony methods have assumptions, and you have to be aware of in your paper (the comparison in ref. 21 was flawed for this reason).

Response: We thank the reviewer for pointing out these. We now cite the additional references in the revised manuscript. The revised manuscript also includes additional comparisons (based on benchmark *C. elegans* datasets) of LinTIMaT with neighbor joining method for reconstructing trees. As for the proposed comparison with the computational method of Ref 21 (developed for analyzing the datasets from the LINNAEUS experimental method), as suggested by the reviewer we attempted to perform such comparison. We downloaded their code and attempted to run it (initially on their own data and then apply it to the data we analyze). Unfortunately, their code does not work. We emailed the authors of the paper for help, but they were unable to provide any working code despite several repeated requests (we cannot even compile the code they provide). Given these issues we were unable to perform the requested comparisons.

R1.C7

It would be worth mentioning in the discussion that the approach of refining a lineage with cell transcriptomes has a potential downfall. It is currently unclear if cell trajectories inferred by transcriptional state and lineage should be concordant in all cases, and although this method likely improves trees on the whole, it may obscure places where the cell lineages have divergent cell types late in development.

Response: While we agree with this comment, and mention it now in Discussion as suggested, we note that for the revised version, we performed additional experiments to assess LinTIMaT's ability to accurately reconstruct lineage relationships in cases where transcriptional state and lineage are discordant (please see response to R2.C1a for details). As we now show in Supplementary Figures S6-S8, even in such cases LinTIMaT is able to accurately reconstruct the lineage branching. Specifically, as we now discuss in Results, LinTIMaT does not override the mutation information based on the expression data. It simply uses the expression data when the mutation data is more ambiguous (due to noise, dropouts etc.). Thus, under reasonable assumptions about such noise, even if the case the reviewer mentions holds, LinTIMaT can still reconstruct an accurate representation of the lineage tree.

Minor points:

1. Line numbers are very helpful in review (for next time)

Response: We have added line numbers in the revised manuscript.

2. There is some use of imprecise language throughout the paper. For instance, "LinTIMaT achieved better accuracy in lineage reconstruction compared to that of MP (for 0.2 they are almost the same, though for lower rates LinTIMaT is significantly better)." It's fine to say that LinTIMaT scored worse

here, and again to the major point, worth explaining. The language of ‘significant’ and ‘almost the same’ should be cleaned up, as people would expect significant in this context to be statistical significance.

Response: We have carefully removed the imprecise language in the revised manuscript.

3. When you analyzed the CRISPR barcodes with variable editing rates (midway through page 6), does LinTIMaT provide an estimate of the editing rate at each target? Was it accurate?

Response: LinTIMaT does not provide an estimate of the editing rate at each target. Instead, it relies on a probabilistic model which does not explicitly model these rates at each target. We now mention this in Methods.

4. Include references to the figures for all the examples. For instance, when you talk about her4.1+ positive (figure 3, not mentioned), it took me a minute to figure out what figure you meant, as the text just walked through figure 4.

Response: We apologize for the inconvenience the reviewer had to face. In the revised manuscript, we have included references to the figures for all the examples.

Reviewer #2

R2.C1

The recurring fundamental problem that we find with this study is that it fails to clearly define the model of the biology that the algorithm is trying to solve for. It also (as a result) lacks a critical discussion of where this model would apply and where it might fail. This lack of clarity occurs both in defining the initial optimization problem, and then again in consensus tree-building. From reading the paper, we can guess what the model being proposed is: namely, that development encodes a tree of molecular states, with a strict and unambiguous increase in the distance between branches at each step of differentiation (e.g. bifurcation). (By “unambiguous”, we mean that the many decisions made in clustering scRNA-Seq data will correctly find this distance, and that it increases monotonically). Further, that lineages invariantly reflect the same tree of molecular states. This description is likely correct for much of C. Elegans development, e.g. in 16 cell embryo used as a benchmark here.

Response: We agree that the models we attempted to optimize were not clearly spelled out. In the revised version, we have explicitly defined the generative model we assume for both, organism specific tree and an invariant species tree. Note however, that it is *not* our intention to model development with this method. Our goal is to model the likely set of events that occurred as part of the CRISPR barcoding experiments. Interpreting these events remains a major challenge but is outside the scope of our paper. Our goal is to simply reconstruct the most likely set of events that led to the set of cells profiled, which is exactly the goal of the Camin-Sokal MP method as used in scGESTALT. This is now explicitly mentioned in the introduction.

For the organism-specific lineage, to determine the most likely set of events we combine both CRISPR mutation data and gene expression data. This is indeed different from prior methods and we believe helps improve the reconstructed tree. However, as our model clearly indicates the use of expression is only to complement and not to override the mutation information. In other words, our generative model allows for very different expression patterns in neighboring branches and very similar expression patterns in very distant branches.

The generative model is described in Methods and Results and illustrated in a new Supplementary Figure, S1. As mentioned above, we use the Camin-Sokal MP method for modeling the CRISPR-Cas9 edits.

Basically, we assume that the edits are acquired on the branches of the cell lineage tree as the single-cell zygote transforms into an adult organism during the course of the lineage recording experiment, as was done by scGESTALT. As for gene expression data, we assume that the expression of a set of cells in a subtree rooted at an internal node of the lineage follows Gaussian distribution. However, cells under each internal node can either have a similar expression profile (which would result from low variance for the Gaussian) or different expression profiles (resulting from high variance for that node). In the latter case this can be the result of two or more different Gaussians (two or more different expression patterns) for descendants of that subtree. To model this, we use a well-known computational framework based on Bayesian Hierarchical Clustering. The key issue for consistency with the mutation data is the weighting of the two types of information we use (mutations and gene expression). To make sure expression is only used in cases where the mutation data is ambiguous (for example due to noise or dropout) we apply a very strict threshold to the mutation likelihood so cells with very different mutations cannot be close in the tree. We now demonstrate this in additional analysis we performed as we discuss in response to another comment below.

For the species invariant tree (previously consensus tree), based on this and comments below we have made changes to our method that result in the following model. Our model assumes that a subset of the lineages (and cells) are conserved between different individuals of the same species. Our method attempts to identify such groups of cells based on both, their similar expression pattern and their branching history. The invariant tree inferred by our method preserves the organism-specific relationships between the clusters while still combining clusters across individual samples. Similar to the organism-specific tree, we describe the model for the invariant tree in Methods and Results (see also Figures 4 and 5).

However, this approach is problematic to generalize for at least two reasons:

a. First, some mature cell types can appear similar but have very distinct developmental relationships, or different but have close ancestry (“convergence” and “divergence” summarizing the two violations of what we have guessed is the model). As a case in point, C. Elegans neurons will tend to co-cluster but diverge very early on in their lineage. They will differ in important ways, but be similar in their major program. In such a case, it is not clear that gene expression distance will correctly separate by lineage. Such “convergence” in aspects of cell state directly pits gene expression similarity against lineage similarity, and it could be wrong to assume that transcriptionally-similar cells have close lineage relationships. As another case, cells at different spatial regions, well-separated by lineage, can have very similar transcriptomes. In a third example, in the ScarTrace paper it was found that macrophages in the tail of zebrafish can have different lineage origins but appear similar by transcriptional clustering. And so on.

Response: As noted above, our method is primarily driven by the sequence (CRISPR) data and only uses expression to correct errors related to CRISPR mutations. To test whether our method can handle the scenarios discussed above, we performed additional simulation experiments based on each of the specific cases (“divergence” and “convergence”) the reviewers raise. To conduct these simulation experiments, we combined experimental zebrafish scRNA-seq (from scGESTALT) with simulated CRISPR-Cas9 mutation data as we have done in the original submission.

The first simulation experiment (“divergence” case) evaluates LinTIMaT’s ability to correctly infer the lineage for two groups of cells that are transcriptionally very similar (same cell type) but diverged very early on in their lineage. We selected forebrain neuron cells from hypothalamus (cell type 27 in ZF1 dataset of scGESTALT) and divided them into two groups (containing 10 and 11 cells respectively). In the simulated lineages (consisting of 100 cells), these two groups were placed in two different subtrees. We simulated lineages under two different settings – 1) two groups diverged directly from the root and did not share any CRISPR mutation (see Supplementary Fig. S6a for an example), 2) two groups diverged

very early on (child of root node being their most recent common ancestor) and possibly shared some CRISPR mutation (see Supplementary Fig. S7a for an example). For both types of lineages, we simulated CRISPR mutations under different experimental conditions (different mutation rates and dropouts, see Results for details) and tested whether LinTIMaT was able to infer the divergent lineage relationship between these two groups despite them being transcriptionally similar. As we show, for such expression divergence under different experimental conditions, LinTIMaT correctly recovered the divergent relationships and placed them in different subtrees (see Results, Supplementary Figs. S6 and S7 for details).

In the second experiment (“convergence” case), we evaluated LinTIMaT’s ability to correctly infer the lineage for two groups of cells that are transcriptionally very different (completely different cell type) but share the same ancestral relationship. We selected two groups of cells, the first group consisting of forebrain neuron cells from hypothalamus and the second group consisting of progenitor cells (from ZF1 dataset of scGESTALT). In the simulated lineages (consisting of 100 cells), these two groups were placed next to each other (see Supplementary Fig. S8a for an example). We simulated CRISPR mutations on the simulated lineages under different experimental conditions (different mutation rates and dropouts, see Results for details) and tested whether LinTIMaT was able to infer the convergent lineage relationship between these two groups despite them being transcriptionally very different. Again, as the figures show, under different experimental conditions, LinTIMaT correctly recovered their convergent lineage relationships and placed them in the same subtree (see Results, Supplementary Fig. S8 for details).

These new experiments illustrate that even though LinTIMaT uses expression data for reconstructing lineage, it does not override the mutational signal when such signal is clear. Thus, LinTIMaT can recover both divergent and convergent lineage relationship based on the available mutational signal.

b. Second, most tissues and animals do not develop through an invariant coupling of molecular state and division history in a manner stereotyped by C. Elegans. In such a case, there are certainly strong rules constraining the distribution of lineage relationships between cell types, but the notion of a “consensus tree” would need to be clearly defined. Does this describe the “most likely tree”, or “average tree” (and if so, how does one understand the idea of an “average”?), or does it seek to capture a core “invariant” topology that is true in all cases, while discarding fate choices that are not invariant? Each of these cases discards some aspect of the real biology. Models do not need to be correct of course – they need to be useful. But the practical uses of these trees is not discussed, or the risks of their misinterpretation. The goal of “consensus tree” building sounds potentially valuable, but also completely unclear.

Response: As the reviewer suggested we do not use the term ‘consensus’ anymore and instead focus on the invariant lineage. The main difference is that unlike in our original version we do not require that all clusters in all individuals need to be included in the invariant lineage. Instead, we now start with a filtering step that identifies clusters that are preserved in all individuals of the same species (based on expression similarity). This step addresses several of the comments the reviewer raised including issues related to organism specific clusters, parameter estimation, down-sampling etc. While this reduces the number of clusters that are included in the invariant tree (by 30-60% depending on the dataset) it leads to a much more robust outcome focusing only on those clusters that are conserved both in terms of their lineage splits and in terms of expression. Detailed model for this version is presented in Methods. The Results are presented in Figure 4, Figure 5, Supplementary Figures S17- S20 and Supplementary Tables S10-S11. As can be seen, while the tree includes fewer clusters compared to the individual trees, biologically the results are comparable to the version presented in our original submission. Specifically, as before, we observed spatial enrichment for the invariant clusters in all three regions of brain (hindbrain, forebrain and midbrain) (Fig. 5a). We also analyzed the functional significance of the joint clusters included in the invariant lineage tree by performing Gene Ontology (GO) analysis. For the scGESTALT

datasets, the invariant clusters showed coherent enrichment of GO terms for all three major cell types (neuron, blood and progenitor) (Fig. 5b). Similarly, for ScarTrace datasets, three major cell types (immune, eye, neuron) were all enriched in invariant clusters (Supplementary Fig. S19). Importantly, the invariant lineage clusters improved on the individual clusters by uncovering more GO terms with more significant (lower) p-values compared to the individual lineage clusters (Fig. 5b, Supplementary Fig. S20). In addition, invariant clusters also correctly uncovered some GO functions that were not enriched in individual clusters including myeloid cell development (p-value < 0.035), neurotransmitter release cycle (p-value < 0.014) for scGESTALT and leukocyte differentiation and migration (p-value < 0.048) and photoreceptor cell outer segment organization (p-value < 0.0003 for ScarTrace (Supplementary Tables S10-S11).

We view these questions as crucial, particularly because lineage-tracing is held as one of the most powerful tools of developmental and stem cell biology, used to definitively “prove” relationships. Historically, the utility of lineage-tracing has been in that it seeks to provide very clean and interpretable results. If an algorithm stands a risk of generating artifacts, or obscuring genuine heterogeneity, this could be problematic if the assumptions being made cannot be simply and clearly formulated for criticism.

*We suspect that these criticisms can be addressed by the authors carrying out two tasks: (a) clearly defining their “toy model” of development, and discussing when it might *not* hold; and (b) demonstrating how the model performs under conditions when the assumptions are maintained *and* when they are violated, namely: ‘convergent’ differentiation (such as the neuronal example above), or non-invariant lineages (e.g. when two or more tree structures co-exist, e.g. in different parts of the embryo). The paper will ultimately be much more useful to biologists if they know what the algorithm cannot do, as well as what it does. Conversely, if they fail to do this they may risk having developmental biologists use their algorithm and then make strong claims that could end up being artifactual.*

Response: As mentioned above, we have performed both. We now explicitly define our generative models and what they do and do not assume. We have also performed the experiments mentioned by the reviewers and show that even when assumptions about expression similarity and dissimilarity do not hold uniformly our method can still recover divergent and convergent lineage relationships between groups of cells.

R2.C2

Even for the toy model proposed here, how well the current algorithm actually works remains unclear.

R2.C2a

The key problem is that there is no underlying truth for the real datasets considered. Instead, the authors mostly focus on analyzing the coherence of their reconstructed lineage tree within each cluster. But the logic is clearly circular: they enforce the tree to have coherent cell types within each cluster in the first place by combining the clonal and transcriptome data, and then they use cell-type coherence to evaluate how successful their approach is. Their metric shows that they manage to minimize their error function. Well done. But it does not generally prove that they manage to correctly construct lineages. Their sole “true” test of the model uses toy data from a C elegans division tree, which as discussed above may not be generalizable even to later stages of C elegans development.

Response: We agree with this comment. Indeed, for the zebrafish data there is no ground truth and so the only way to evaluate our results is to use an external information source and compare agreement on that between different methods. Some of this is indeed circular and we now explicitly acknowledge it in the paper (specifically the agreement with the clustering). Other is more indirect. We also compared the lineage trees reconstructed by the methods using GO analysis. Unlike the clustering we compare to GO

information was obtained independently from the data we use and so is a fair way to compare results. The clusters in the LinTIMaT reconstructed lineage uncovered more GO functions compared to the barcode clusters of Camin-Sokal MP lineage (see Results and Supplementary Fig. S13 for details). In addition, we use GO analysis to show that the invariant tree obtains better results than the individual trees obtained by all methods. Finally, we use ground truth data in our simulation analysis which clearly shows the advantage of using expression data both when the assumptions we make uniformly hold and in cases where they do not hold for some of the branching / split events. We believe that combined these analyses prove that LinTIMaT can improve the reconstruction of the set of events that occur during the experiment which is the ultimate goal.

R2.C2b

*Even for their benchmarking with *C elegans*, the analysis should be improved: the authors picked a very simple 2-division model to test their algorithm, and they compare their algorithm to a null method that involves Maximum Parsimony without considering Camin-Sokal criteria. It would be much more instructive (and fair) to compare their algorithm to Maximum Camin-Sokal Parsimony, as this method is well-adopted in the field. They should also compare to other methods, like neighbor joining etc. Essentially, the current test doesn't reveal whether there is a major improvement from scRNA-Seq information, even in this simple toy example. To really assess whether transcriptome information helps to improve the tree construction in this context, it would be interesting to compare directly, with and without this transcriptome information, how their maximum likelihood approach works.*

Response: We have performed the analyses suggested and results are presented in Figure 2 and Supplementary Figures S2-S5. Comparison to the *Maximum Camin-Sokal Parsimony* method the reviewer proposed was actually already performed in our original submission, but we mistakenly did not use the correct term for this. We have now also added a comparison with neighbor joining as suggested. As the figures show, LinTIMaT improves over both methods in all settings we looked at. We also present new comparison results corresponding to a range of mutation rates for the dropout experiment (Supplementary Figure S5). As we show LinTIMaT is still the best even when considering different possible rates for these mutations.

As for comparison of our method with and without expression data. Our likelihood formulation for the CRISPR mutations essentially gives us likelihood values that are similar to the most parsimonious (under Camin-Sokal criteria) assignment of the mutation states at the internal nodes of the lineage. Thus, the comparison we present to the Camin-Sokal maximum parsimony method is essentially addressing the request to compare our method with and without the use of expression data.

R2.C2c

(As noted above, it will be instructive to know under what realistic conditions their method does not accurately reconstruct lineage structure, or rather what aspects of lineage structure does their method really capture when invariant lineage hierarchy does not exist).

Response: Based on the previous comments we now limit the analysis of the invariant tree to clusters that are similar in expression above a certain threshold. This would mean that the combined tree may leave out several key clusters (or lineages) if their expression levels are not well conserved between different organisms from the same species. We now mention this in Discussion as a potential shortcoming of the method.

R2.C3

Some technical issues regarding consensus clustering:

• *There is a strong assumption here that there are a common set of clusters, defined here to be the maximum number of clusters from one of the trees. Could this become violated in some realistic applications, making the algorithm fragile? Could their method deal with a case where one individual has few clusters, but one of these few cluster is not found in all other individuals?*

Response: As mentioned above, based on the reviewer comments we have now changed the initial assignment algorithm. In the new algorithm we only use clusters that match well with clusters in another organism (sample). Thus, we no longer require that the maximum number of clusters from one of the trees be used.

• *The approach itself (as described in methods) appears fragile in some aspects: the authors need to balance the number of cells for different individuals; they need to tweak the number of cell type clusters for individuals (Page 10 in main text, “we subsampled 380 cells for ZF1 so that both fish have equal weights when learning the consensus tree. LinTIMaT inferred 43 clusters for ZF1 and 42 clusters for ZF3 and so we split one cluster in ZF3 to obtain 43 clusters for both trees”), etc.*

Response: We agree. This is one of the main reasons we now changed the initial assignment method. In the new version (described in Methods) we have a cutoff on both the similarity of a match and the number of cells required in each of the constituent clusters to make sure that these are indeed invariant clusters. This reduces the need to tweak the algorithm when applying to new data. The user still needs to select these thresholds (on the similarity and number of cells) but we believe these are more intuitive for users as opposed to subsampling for example.

• *Their choice of ARI to assess the performance of their consensus clustering (Fig. S10) is again circular. Can they consider a non-circular way of quantifying how well their method performs?*

Response: As mentioned above, we agree that using ARI, which quantifies the agreement with the expression-based clustering, leads to a circular reasoning. We mentioned a few additional ways we are trying to validate the results. These include GO analysis and simulation in which we know the ground truth. As the reviewer noted, there is no ground truth for the zebrafish datasets and so we resort to both independent (GO) and related (cluster ARI) analysis.

Minor comments:

• *As a suggestion, some similar works are now on bioarxiv and it may be helpful to comment provide on these related work. e.g., J. Feng et al., Estimation of cell lineage trees by maximum-likelihood phylogenetics (2019), M. G. Jones et al. Inference of Single-Cell Phylogenies from Lineage Tracing Data (2019). Since these papers are not past peer review, this is at the authors’ discretion. For a reader it would no doubt be valuable to know the differences between the methods.*

Response: We thank the reviewers for mentioning these similar works. We now cite them in the revised manuscript.

• *A better explanation for ARI might be helpful. Does the absolute value of ARI has any meaning? Why it is so low, e.g., 0.07?*

Response: ARI compares the overlap between two different clustering results taking into account both the number of clusters and the number of samples being clustered. The ARI ranges from -1 to 1 with 0 meaning random overlap.

As for the specific ARI values we obtain, we note that we used ARI to show that improving our target function also improves the ARI. So, our focus was not on the absolute agreement with the expression only clusters but rather on improvement to this agreement as part of our iterative learning process. As for the question about the value, the values we obtain are low is due to the constraint imposed by the CRISPR mutation information. As we showed above, our method does not override this information whereas the clustering in the original scGESTALT paper only used expression. Note also that the clustering in the paper itself may not be accurate in terms of cell type assignment since it is itself based on manual curation of a t-SNE reduced dimension clustering. Other issues leading to the low ARI are the large number of clusters, size distribution for the clusters, the fact that they are based on non-linear t-SNE projections, and the facts that boundaries are often manually set. In addition, the original clustering (63 cell types) was determined based on data integrated from 6 zebrafishes and their distribution in each individual fish might vary making it even more difficult to represent them from the lineage tree derived clusters that are constructed based on tree structure defined by both CRISPR mutations and gene expression.

- *A typo in Eq. 10. The last term, $du*dv$ should be $du*dw$.*

Response: We have fixed this typo in the revised manuscript.

- *Could the authors please comment on the computation time?*

Response: In the Supplementary Results section of the revised version, we have added a subsection (2.5) which presents the runtime of LinTIMaT for the various datasets we analyzed.

REVIEWERS' COMMENTS:

Reviewer #1 (Remarks to the Author):

Zafar and Lin review.

Overall I think the authors did a nice job addressing reviewer comments. The suggested convergence and divergence experiments are a nice addition to the paper, and I appreciate the updates to the methods and thank the authors for trying to get the software for ref 21 running. Saying that, I am a little worried about the new 'invariant lineage trees' terminology. The shared lineages of any set of sequenced samples (say the 3 zebrafish) doesn't necessarily mean the resulting lineages are invariant, only that the captured patterns are shared within these samples. Invariant has also long been used to describe organisms with wholly invariant lineages such as *C. elegans*. I know it was taken on to meet the criticisms in the first round of review, but I worry the community at large will have issues with this. In the end, I'll leave it up to the editor.

-Aaron

Minor text points I saw:

Line 7: "Such studies resorted to.." Something like leveraged or another more positive word here.

Line 74 "display a similar expression profile", add the a

The section title on line 286 could use rephrasing unless the method is called 'Invariant lineage tree'

Reviewer #2 (Remarks to the Author):

The revised manuscript is much improved, and has successfully addressed our concerns.

The following minor issues could be fixed:

1, In response to our comments, the authors now introduce the terms "convergent/divergent differentiation". Their definition and subsequent usage of the terms (lines 163 onwards) is opposite to what we are familiar with (Wagner et al, Science 2018 and Klein and Wagner, NRG 2020), and we would argue that their definition is less clear than ones we have seen before. We recommend they flip their definitions. In particular, we recommend that they define the terms with respect to the final transcriptome state, rather than the (unobserved) ancestral state.

For example, here, the authors describe "convergence" as "cells from different cell types can share close ancestry" – this is only "convergent" if one walks backwards in time so that distinct cells "converge" in lineage. The definition we have previously seen (and used) is that "convergent differentiation" is a case when lineage distance is larger than expected given the transcriptional similarity of cells. That is, cells appear to converge to the same transcriptomic state despite being more distant in division history than expected. A tree defined strictly based on transcriptional similarity would not faithfully reflect division history because cell that appear close on such a tree turn out to be far apart on a division tree. "Divergent differentiation" means the opposite: cells that are close in division lineage diverge to quite different transcriptome states in the end.

2, Regarding the simulated test on convergent differentiation (Fig.S6-S7), the authors have used very distinct lineages, and have been able to recover the lineage with a high success rate. As we understand it, it is simply because the lineage history is so different that the similarity of transcriptome state no longer matters. In their algorithm (Eq.(12), line 520 in Method), there is a tuning parameter ω_2/ω_1 which controls the relative contribution of the transcriptomic information in the construction of the lineage tree. When this ratio is small enough, the outcome will be primarily determined by the lineage history; however, at the same time, the

real benefit of applying this method, i.e., leveraging the transcriptomic information for lineage reconstruction, is also seriously limited. In the end, it is a matter of how informative the lineage data is, and how one trusts the transcriptomic information in helping to reconstruct the lineage. This answer probably varies among different experimental systems, and may change with the improvement of lineage tracing technology. While the author has already acknowledged in the discussion that "it is still possible that in some cases the use of expression data may lead to less accurate reconstructions" (line 400), they should be more explicit about the reasons of success or failure, in the section of discussion, and also in the section of the simulated test on convergent/divergent differentiation. This will provide better guidance for others in using this method for interpreting their data.

3, The method section on finding invariant lineage is still very difficult to follow. Importantly, at line 600, " For each pair of ..." It is unclear to me what the pair is. I assume the authors are talking generally about finding invariant lineage among several individual lineages. (I think the authors have in mind their zebrafish data, which has only two individuals). Another example is at line 604-606, especially "...if none of the clusters are matched,..." ". I simply could not follow this sentence, although I can vaguely guess the message. I would suggest the authors to carefully rewrite this part to make it more transparent.

4, The authors should clarify how their method deals with the case of dropout. Is it modeled as an un-mutated state? How does this event change the mutation fraction m_s in Eq. (5) of Methods? Dropout can be a correlated event that involves several target sites, and therefore cannot be described by their simple site-independent transition model in Eq. (5) of Methods.

5, Change "consensus lineage" in Fig. 1 to "Species invariant lineage" for consistency

Best,

Shou-Wen Wang and Allon Klein

Point-by-Point Responses for Reviewers

NCOMMS-19-29009A

Single-cell Lineage Tracing by Integrating CRISPR-Cas9 Mutations with Transcriptomic Data

Hamim Zafar; Chieh Lin; Ziv Bar-Joseph

Dear Editors,

We detail below the changes we made in response to the remaining minor comments from the reviewers. Updated text is also highlighted in the manuscript for your convenience.

Reviewer #1

Remarks to the Author: Overall, I think the authors did a nice job addressing reviewer comments. The suggested convergence and divergence experiments are a nice addition to the paper, and I appreciate the updates to the methods and thank the authors for trying to get the software for ref 21 running. Saying that, I am a little worried about the new ‘invariant lineage trees’ terminology. The shared lineages of any set of sequenced samples (say the 3 zebrafish) doesn’t necessarily mean the resulting lineages are invariant, only that the captured patterns are shared within these samples. Invariant has also long been used to describe organisms with wholly invariant lineages such as *C. elegans*. I know it was taken on to meet the criticisms in the first round of review, but I worry the community at large will have issues with this. In the end, I’ll leave it up to the editor.

Response: As the reviewer correctly mentions, we moved from using ‘consensus’ to ‘invariant tree’ based on comments by Reviewer 2. We do think that invariant, the way we use here, is stronger than ‘shared’ since these are lineages identified in multiple samples and so are likely those most constrained. We would be happy to hear what the editor thinks as we try to balance the requests of the two reviewers.

Minor points:

Line 7: "Such studies resorted to.." Something like leveraged or another more positive word here.

Line 74 “display a similar expression profile”, add the a

The section title on line 286 could use rephrasing unless the method is called ‘Invariant lineage tree’

Response: We have made these changes in the main manuscript.

Reviewer #2

R2.C1

In response to our comments, the authors now introduce the terms “convergent/divergent differentiation”. Their definition and subsequent usage of the terms (lines 163 onwards) is opposite to what we are familiar with (Wagner et al, Science 2018 and Klein and Wagner, NRG 2020), and we would argue that their definition is less clear than ones we have seen before. We recommend they flip their definitions. In particular, we recommend that they define the terms with respect to the final transcriptome state, rather than the (unobserved) ancestral state.

For example, here, the authors describe “convergence” as “cells from different cell types can share close ancestry” – this is only “convergent” if one walks backwards in time so that distinct cells “converge” in lineage. The definition we have previously seen (and used) is that “convergent differentiation” is a case when lineage distance is larger than expected given the transcriptional similarity of cells. That is, cells appear to converge to the same transcriptomic state despite being more distant in division history than

expected. A tree defined strictly based on transcriptional similarity would not faithfully reflect division history because cell that appear close on such a tree turn out to be far apart on a division tree.

“Divergent differentiation” means the opposite: cells that are close in division lineage diverge to quite different transcriptome states in the end.

Response: We agree with the reviewer. While previously we used the terms “convergence” and “divergence” to denote how close or far two groups of cells are in the lineage (thus the cells were convergent or divergent with respect to an unobserved ancestor), as the reviewer suggests it makes more sense to define these terms with respect to the final transcriptome state of the cells to be consistent with current literature (Klein and Wagner, NRG 2020). In the revised manuscript, we have thus redefined these terms with respect to the final transcriptome state. In particular, now in our manuscript “convergent differentiation” refers to the scenario where cells having distinct developmental history converge onto the same mature cell type, whereas “divergent differentiation” denotes the scenario where cells sharing close ancestry differentiates into distinct transcriptional states. In order to maintain consistency with this new nomenclature, we have appropriately updated the subsections in Results and Supplementary Discussion that describe the corresponding experiments. Also, the captions of the Supplementary Figures 6-8 that display the results of these experiments have been updated to reflect this new nomenclature.

R2.C2

Regarding the simulated test on convergent differentiation (Fig.S6-S7), the authors have used very distinct lineages, and have been able to recover the lineage with a high success rate. As we understand it, it is simply because the lineage history is so different that the similarity of transcriptome state no longer matters. In their algorithm (Eq.(12), line 520 in Method), there is a tuning parameter ω_2/ω_1 which controls the relative contribution of the transcriptomic information in the construction of the lineage tree. When this ratio is small enough, the outcome will be primarily determined by the lineage history; however, at the same time, the real benefit of applying this method, i.e., leveraging the transcriptomic information for lineage reconstruction, is also seriously limited. In the end, it is a matter of how informative the lineage data is, and how one trusts the transcriptomic information in helping to reconstruct the lineage. This answer probably varies among different experimental systems, and may change with the improvement of lineage tracing technology. While the author has already acknowledged in the discussion that “it is still possible that in some cases the use of expression data may lead to less accurate reconstructions” (line 400), they should be more explicit about the reasons of success or failure, in the section of discussion, and also in the section of the simulated test on convergent/divergent differentiation. This will provide better guidance for others in using this method for interpreting their data.

Response: As suggested, we have added text to Discussion and to Supplementary Discussion to further emphasize the tradeoff the algorithm makes between mutation and expression information. In Discussion we now say ‘Our method contains a user defined parameter for weighting the contribution of the mutation and expression data. While our analysis indicates that the outcome is usually robust to changes of this parameter, a user can change the value of the weight parameter to reflect their belief about the importance of the two types of data we consider.’ In Supplementary Discussion on the convergence differentiation experiment, we now discuss an example for which LinTIMaT’s use of expression data led to less accurate recovery of the lineage relationship between two groups of cells that converged to the same transcriptional state despite having slightly dissimilar ancestry. For this example, LinTIMaT faced challenge as the groups shared a common mutation but the branches that separated the two groups did not have any mutational information because of dropout. It is important to note that any lineage reconstruction method relying on mutation information (Maximum Parsimony or Neighbor-Joining) would have suffered from the same problem in such a scenario. We believe the discussion of this example will be helpful for others using LinTIMaT for interpreting their data. In addition, in the same subsection of Supplementary Discussion, we say ‘Results for this simulation experiment are based on a default

setting for the weight parameter for the expression data. A very low weight for the expression data (compared to the weight for mutation data) may lead to different results that may not accurately reconstruct the lineage.”

R2.C3

The method section on finding invariant lineage is still very difficult to follow. Importantly, at line 600, “For each pair of ...” It is unclear to me what the pair is. I assume the authors are talking generally about finding invariant lineage among several individual lineages. (I think the authors have in mind their zebrafish data, which has only two individuals). Another example is at line 604-606, especially “...if none of the clusters are matched,... “. I simply could not follow this sentence, although I can vaguely guess the message. I would suggest the authors to carefully rewrite this part to make it more transparent.

Response: We have carefully rewritten this part to make it more transparent and easier to follow. We have described how we form candidate cluster groups by selecting one cluster from each individual lineage and how we compute gene expression distance for each such cluster group. We have also elaborated on how we perform the greedy matching that selects the cluster groups to be incorporated in the invariant lineage. Please see Methods (lines 565-580) for more details.

R2.C4

The authors should clarify how their method deals with the case of dropout. Is it modeled as an unmutated state? How does this event change the mutation fraction m_s in Eq. (5) of Methods? Dropout can be a correlated event that involves several target sites, and therefore cannot be described by their simple site-independent transition model in Eq. (5) of Methods.

Response: Our mutation likelihood formulation does not explicitly account for dropouts. So, the answer is yes, if a dropout occurs the state is assumed to be unmutated. Of course, there is no way to distinguish between dropouts and non-mutation events so one can use one or a combination of two strategies. What we are using to address this issue is the expression data. As can be seen from Fig. 2d, in the presence of dropout, the use of expression data helps LinTIMaT in recovering more correct branchings compared to the reconstruction methods that do not account for dropouts at all (MP or NJ). The second strategy uses priors to change the parameters of the models and thus requires specific assumptions. Essentially, dropout of a mutation will result in the observed m_s being lower than the actual mutation fraction. If a user has a good estimate of the dropout rate (which we did not have for this data) it can be incorporated in the transition probability model by modeling m_s as a function of actual mutation fraction and dropout rate, where actual mutation fraction is modeled as a beta distributed random variable whose values are simultaneously estimated while reconstructing the lineage. While we have not used this strategy in the current analysis, if such a prior exists, it should be possible to update the model to utilize it. We have added this information to Methods when discussing the transition probability matrix and its parameters.

R2.C5

Change “consensus lineage” in Fig. 1 to “Species invariant lineage” for consistency

Response: We have made the change to make the terminology consistent.